

# Full-scale spectra of 15-year time series of near-surface horizontal wind speed on the north slope of Mt. Everest

Cunbo Han[1,3], Yaoming Ma[1,2,3,4,6], Weiqiang Ma[1,3], Fanglin Sun[5], Yunshuai Zhang[1,3], Wei Hu[1,3], Hanying Xu[1,3,4], Chunhui Duan[1,3,4], Zhenhua Xi[1,3]

[1]Land-Atmosphere Interaction and its Climatic Effects Group, State Key Laboratory of Tibetan Plateau Earth System, Resources and Environment (TPESRE), Institute of Tibetan Plateau Research, Chinese Academy of Sciences, Beijing 100101, China.
[2]College of Hydraulic & Environmental Engineering, China Three Gorges University, Yichang 443002, China.
[3]National Observation and Research Station for Qomolongma Special Atmospheric Processes and Environmental Changes, Dingri 858200, China.
[4]University of Chinese Academy of Sciences, Beijing, 100049, China.
[5]Key Laboratory of Land Surface Process and Climate Change in Cold and Arid Regions, Northwest Institute of Eco-Environment and Resources, Chinese Academy of Sciences, Lanzhou 730000, China
[6]College of Atmospheric Sciences, Lanzhou University, Lanzhou 730000, China

*Correspondence to*: Cunbo Han (cunbo.han@hotmail.com) and Yaoming Ma (ymma@itpcas.ac.cn)

**Abstract.** Wind speed spectra analysis is of great importance for understanding boundary layer turbulence characteristics, atmospheric numerical model development, and wind energy assessment. 15-year time series of near-surface horizontal wind data from the national Observation and Research Station for Qomolongma Special Atmospheric Processes and Environmental Changes (QOMS) on the north slope of Mt. Everest has been used to investigate the full-scale wind spectrum
in the frequency range from about 10 yr$^{-1}$ to 5 Hz. The annual average wind speed showed almost no detectable trend from 2006 to 2018 at the QOMS station. Three peaks were identified in the full-scale spectra at the frequencies of 1 yr$^{-1}$, 1 day$^{-1}$, and 12 hr$^{-1}$, respectively. The 12 hr$^{-1}$ peak is rarely observed at an offshore, coastal, or terrestrial site, and indicates the unique local circulations at the QOMS station. The spectral density was the highest on the low-frequency side of the diurnal peak and in the microscale frequency range ($f \geq 1\times10^{-3}$ Hz) in winter, indicating frequent synoptic weather events and
vigorous turbulent intensity generated by shear due to strong wind during winter. An obvious spectral gap around the frequency of $4.5\times10^{-4}$ Hz was observed in the composite seasonal and daily spectrum in winter, while the spectral gap disappeared in summer. The linear composition of microscale and mesoscale wind spectra also held, and the gap region of the horizontal wind spectrum was modeled very well at the QOMS site.

## 1 Introduction

Mt. Everest (Qomolangma), the highest mountain on Earth, is located in the central part of the Himalayas on the southern edge of the Tibetan Plateau (TP). The Mt. Everest region is the frontline for experiencing global climate change and has



unique responses to climate change different from other places of the TP (Kang et al., 2022; Ma et al., 2023), and has experienced significant warming, substantial decrease in precipitation, glacier and snow retreat, and permafrost degradation over the past decades (Yang et al., 2014; Salerno et al., 2015; Xu et al., 2016; Han et al., 2021; Kang et al., 2022; Ma et al.,

2023). The changes affect local and regional atmospheric circulations, especially the local land-atmosphere interactions. To monitor the atmospheric and environmental conditions, a comprehensive observation network has been established in the north slope of Mt. Everest since 2005 (Ma et al., 2023). Based on the observation data, land surface processes, atmospheric boundary processes, atmospheric aerosols and pollutants have been intensively studied (Chen et al., 2013; Han et al., 2015; Sun et al., 2018; Lai et al., 2021; Kang et al., 2022; Ma et al., 2023).


Wind speed spectra, referring to the frequency distribution of wind speed energy within the atmosphere, are of great importance to analyze whether the wind speed energy is evenly distributed over different frequencies or concentrated in a certain frequency range. The spectra are fundamental in understanding the characteristics of atmospheric flow, turbulent transport processes, and wind energy distribution (Van der Hoven, 1957; Fiedler and Panofsky, 1970; Larsén et al., 2013;

Kang and Won, 2016; Larsén et al., 2016; Gyatso et al., 2023). In practice, wind spectral analysis is widely used in understanding the dynamics of atmospheric boundary layers (Zou et al., 2020; Larsén et al., 2021; Li et al., 2021; Lin et al., 2021; Williams and Qiu, 2022), validating atmospheric numerical models (Dipankar et al., 2015; Schalkwijk et al., 2015), and wind energy resource assessment (Escalante Soberanis and Mérida, 2015; Watson, 2019; Effenberger et al., 2024; Liu et al., 2024).


There is a long history of studies on the spectral characteristics of atmospheric boundary-layer horizontal wind velocity. Van der Hoven (1957) analyzed a full-scale spectrum of horizontal wind speeds measured at 91 m, 108 m, and 125 m heights at the Brookhaven National Laboratory. The wind speed spectrum has two majority peaks: one located at a period of 4 days and a second peak occurring at a period of about 1 minute, corresponding to the synoptic-scale and turbulent motions,

respectively. Between the two major peaks is a wide spectral gap at a frequency ranging from 2 h to 10 min due to lack of physical processes that could support wind speed fluctuations in the frequency range. This is the so-called "gap" at a period of about 1 h separating the three-dimensional (3D) microscale turbulence from the two-dimensional (2D) mesoscale to macroscale motions.

There are a lot of discussions on the existence of the spectral gap. Many studies observed the a near-surface gap at periods of about 1 h and confirmed its existence (Fiedler and Panofsky, 1970; Smedman-Högström and Högström, 1975; Gomes and Vickery, 1977; Kaimal and Finnigan, 1994; Vickers and Mahrt, 2003; Kang and Won, 2016; Larsén et al., 2016; Li et al., 2021), although the gap was not as significant as that of Van der Hoven (1957). However, some other studies argued that the

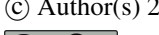



gap does not exist at the period of about 1 h, as it is filled with enhanced mesoscale wind fluctuations due to rolls, jets, and
convective cells (LeMone, 1976; Heggem et al., 1998; Lenschow and Sun, 2007). Studies question the existence of the gap
in Van der Hoven (1957)'s wind speed spectrum mainly because the high-frequency region was observed during the passage
of a hurricane, which increases the spectral density in the high-frequency range compared to standard atmospheric conditions
(Smedman-Högström and Högström, 1975; Lenschow and Sun, 2007; Kang and Won, 2016; Larsén et al., 2016). Larsén et
al. (2016) suggested that the gap is jointly regulated by the variation in horizontal wind variations from the two-dimensional
mesoscale motions and three-dimensional boundary layer turbulence, and may be visible or invisible depending on the
relative contribution to the fluctuation from the microscale and mesoscale motions. The gap also suggests that microscale
and mesoscale motions are weakly correlated or uncorrelated and the atmospheric boundary layer flow could be decomposed
into low-pass filtered large-scale flows and high-pass filtered turbulence (Stull, 1988). Most atmospheric boundary layer
parameterization schemes adopt this concept, explicitly resolving large-scale motions in the numerical simulations while
parameterizing small-scale turbulent motions.

The characteristics of the wind speed spectrum are influenced by various factors, such as topography, land surface conditions,
environment, and observations. Therefore, wind speed spectra at different locations and heights have been intensively
investigated and compared (Wieringa, 1989; Högström et al., 2002; Larsén et al., 2013; Kang and Won, 2016; Larsén et al.,
2016; Li et al., 2021; Lin et al., 2021; Effenberger et al., 2024). Larsén et al. (2013) analyzed the spectral structure of
mesoscale winds at two offshore sites and found that the climatological wind spectra show universal characteristics
consistent with findings in the literature. Larsén et al. (2016) presented full-scale wind spectra of one coastal and one
offshore site at different heights (10, 80, 100 m) and found that the spectral gap exists but becomes less evident with
increasing height. Kang and Won (2016) investigated the spectra of 5-year time series wind speeds at 10 m and 100 m
heights observed at the eastern slope of the Rocky Mountains in Colorado. They suggested that the spectral gap at periods of
about 1 h is not pronounced, while an obvious diurnal peak exists, which is often absent or less noticeable in the spectra at
coastal sites. Lin et al. (2021) analyzed the wind spectra observed at 8 urban surface meteorological weather stations in Hong
Kong and found that spectra peaks at 1 year, 1 day, and 1/2 day are observed at all stations, and the spectral characteristics
vary with topography and seasons. Li et al. (2021) studied full-scale spectra of multilayer wind time series measured on a
325 m high urban meteorological tower in the northern center of Beijing. The results show that the spectral gap exists in the
lowest two layers, becomes less pronounced, and eventually disappears with increasing height. Moreover, the 1 year and
diurnal peaks were observed in all layers. Therefore, issues such as the existence of the spectral gap and the 1/2-day spectral
peak are still being discussed.



This study aims to advance the knowledge of the full-scale horizontal wind speed spectrum in mountainous regions by using extensive datasets of 10-min horizontal wind data and 3D sonic anemometer wind data observed at the northern slope of Mt. Everest over the past 15 years, from September 7, 2005 to June 12, 2019. Section 2 gives the measurements and wind data, as well as the methodology by which wind data is processed and the wind spectra are calculated and analyzed. Section 3 presents results and discussion, followed by summary and conclusions in section 4.

## 2 Data and methodology

### 2.1 The QOMS observatory

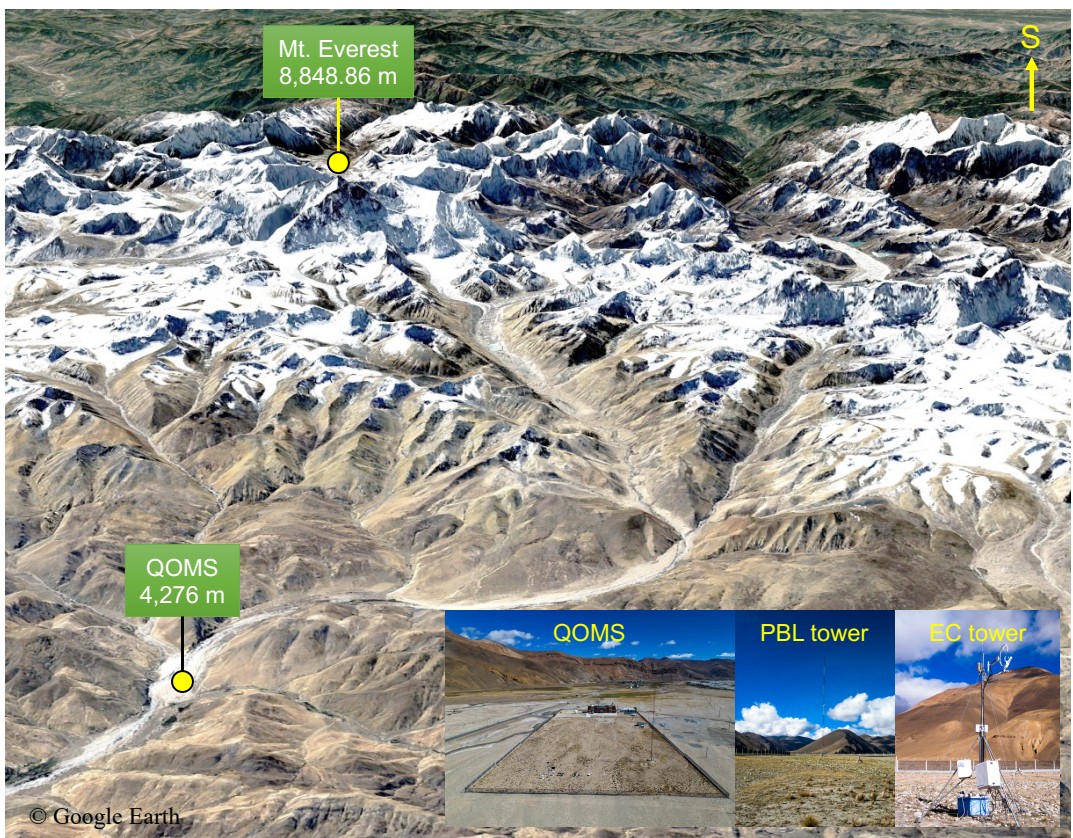

**Figure 1: Instruments (the PBL tower and the EC tower) and landscape around the QOMS station (the terrain map is from Google Earth).**




The national Observation and Research Station for Qomolongma Special Atmospheric Processes and Environmental Changes (QOMS) was established in 2005 and has been gradually developed into a comprehensive observation network including six observation stations in total on the northern slope of Mt. Everest (Ma et al., 2023). The scientific objective of the QOMS is to advance the understanding of land-atmosphere interactions and related climate impacts on the northern slope

of Mt. Everest. The QOMS station is located in the Rongbuk valley, at an elevation of 4276 m above sea level, and is approximately 30 km north of Mt. Everest (Figure 1). The ground surface around QOMS is flat and is covered with sand, gravel, and sparse short grass. A planetary boundary-layer (PBL) tower (Vaisala, MILOS520), an eddy-covariance (EC) tower (Campbell CSAT3 3D sonic anemometer and Li-Cor Li-7500 infrared gas analyzer), and a four components radiation budget system (Kipp & Zonen, CNR-1) have operated since September 2005 and are maintained regularly. The PBL tower is

equipped with five levels of air temperature, specific humidity at heights of 1.5, 2, 4, 10, and 20 m, three levels of wind direction at heights of 1.5, 10, and 20 m, and six layers soil temperature and moisture observations buried at depths of 5, 10, 20, 40, 80, 160 cm. The eddy covariance system is mounted at the height of 3.5 m. Many more instruments have been set up at the QOMS station science 2005, especially after 2019, supported by the second Tibetan Plateau Scientific Expedition and Research Program. More details on the instruments at the QOMS station are found in Ma et al. (2023).

**2.2 Wind data**

We analyzed two types of wind data in this study, 10-min horizontal wind velocity and direction measured with cup anemometers and wind vanes mounted on the PBL tower and 10-Hz 3D wind data observed by sonic anemometer installed on the eddy-covariance tower. The PBL tower and eddy-covariance system have been in operation since September 2005. However, 10-min wind data observed by the PBL tower are not available between June 2019 and December 2019. Thus, 10-

min wind data from September 7, 2005 to June 12, 2019 are analyzed in this study. The 10-m wind data availability is summarized in Table 1 for each year and at the five levels over the 15-year study period. The yearly data coverage of 10-min wind data ranges from 90.1 to 100%, and is generally higher than 95% except for 2008, 2010, and 2013. The 10-Hz 3D wind data were observed at a height of 3.5 m. Data are available from 2005 to 2019. We analyzed the data-available days for each year and found that the data availability was highest in 2015 and 2016. Thus, the 10-Hz 3D wind data from 2015 and 2016

were used to analyze the wind spectra from about $10^{-5}$ to 5 Hz. The 10-Hz 3D wind data availability of each month in 2015 and 2016 is summarized in Table 1 as well. Only days without missing data were selected. All the data used in this study have been published in Ma et al. (2020) and are publicly accessible at the National Tibetan Plateau Data Center (https://doi.org/10.11888/Meteoro.tpdc.270910, last access on June 12, 2024).






**Table 1: Yearly 10-min wind data availability at the five levels on the PBL tower at the QOMS station over the 15-year study period from September 7, 2005 to June 12, 2019, where for each day the data coverage is greater than 99.0%. Available days in each month of the 10-Hz 3D wind data availability at the QOMS station in 2015 and 2016, and for each day the data coverage is 100%.**

| Freq | h (m) | 2005 | 2006 | 2007 | 2008 | 2009 | 2010 | 2011 | 2012 | 2013 | 2014 | 2015 | 2016 | 2017 | 2018 | 2019 |
|---|---|---|---|---|---|---|---|---|---|---|---|---|---|---|---|---|
| 10-min | 1.5 | 99.7 | 99.5 | 99.7 | 90.1 | 98.2 | 54.3 | 99.2 | 99.5 | 90.2 | 95.9 | 100 | 100 | 97.7 | 96.7 | 97.8 |
| | 2 | 99.7 | 99.5 | 99.7 | 90.1 | 98.2 | 91.8 | 99.2 | 99.5 | 90.1 | 95.9 | 100 | 100 | 97.7 | 96.7 | 97.8 |
| | 4 | 99.5 | 99.5 | 99.7 | 90.1 | 98.2 | 91.8 | 99.2 | 99.5 | 90.1 | 95.9 | 100 | 100 | 97.7 | 96.7 | 97.8 |
| | 10 | 99.7 | 99.5 | 99.7 | 90.1 | 98.2 | 91.8 | 99.2 | 99.5 | 90.1 | 95.9 | 100 | 100 | 97.7 | 96.7 | 97.8 |
| | 20 | 99.7 | 99.5 | 99.7 | 90.1 | 98.2 | 44.0 | 0.0 | 6.9 | 90.1 | 95.9 | 100 | 100 | 97.7 | 96.7 | 97.8 |

| Freq | Year | Jan | Feb | Mar | Apr | May | Jun | Jul | Aug | Sep | Oct | Nov | Dec | Total |
|---|---|---|---|---|---|---|---|---|---|---|---|---|---|---|
| 10-Hz | 2015 | 29 | 28 | 29 | 29 | 30 | 30 | 29 | 30 | 29 | 30 | 29 | 30 | 352 |
| | 2016 | 30 | 29 | 30 | 29 | 30 | 29 | 30 | 30 | 29 | 30 | 29 | 30 | 355 |





## 2.3 Method of calculating wind speed spectrum

The power spectra are calculated using the Fourier transform method, and a linear detrending is applied to the wind speed
data time series. As the Fourier transform requires a continuous time series, thus the missing data points in the time series
must be filled. Kang and Won (2016) tested the sensitivity of spectra calculation to the filling methods, including linear,
nearest-neighboring, zero-, first-, second-, and third-order spline interpolation methods, and pointed out that the spectra
results are insensitive to the interpolation method. Thus, in this study, the few missing data in the 15-year 10-min wind time
series were filled in using the linear interpolation method with data before and after the gaps. For the 10-Hz 3D wind data,
only days without missing data points were chosen to calculate spectra for high frequencies.

The power spectral densities are not evenly distributed on a logarithmic scale, sparse in the low-frequency end and dense in
the high-frequency end. Frequency smoothing of the power spectra densities is commonly used to extract a representative
spectral curve from the estimates (Kaimal and Finnigan, 1994). A frequency smoothing method, in which the averaging
interval keeps expanding with frequency, is applied in this study. The number of estimates in each nonoverlapping block
increases exponentially as a function of frequency to yield about seven to eight estimates per decade. In practice, the first
few estimates are accepted as they are, then the number of estimates is increased in steps 3, 5, 7, and so on until the density
of smoothed estimates per decade reaches seven or eight. Details can be found in section 7.4 of Kaimal and Finnigan (1994).

## 3 Results and discussion

### 3.1 Horizontal wind at QOMS

Wind roses calculated from the 10-min wind time series at 1.5, 10, and 20 m heights for the 15-year period are shown in
Figure 2. The compass roses with 32 wind directions show that the southerly and northerly winds are the two predominant
wind directions at the QOMS station. The wind predominantly blows from the north-northeast (13.9%), south-southeast
(12.3%), and south (12.2%) at 1.5 m height. The wind velocity is mainly below 4 m/s. The predominant wind directions are
almost unchanged from 1.5 to 10, and 20 m, north-northeast and south winds are still the prevailing wind directions. While
the wind speed increased significantly with height, a large portion of wind speed is higher than 10 m/s, especially for the
westerly and southerly winds.



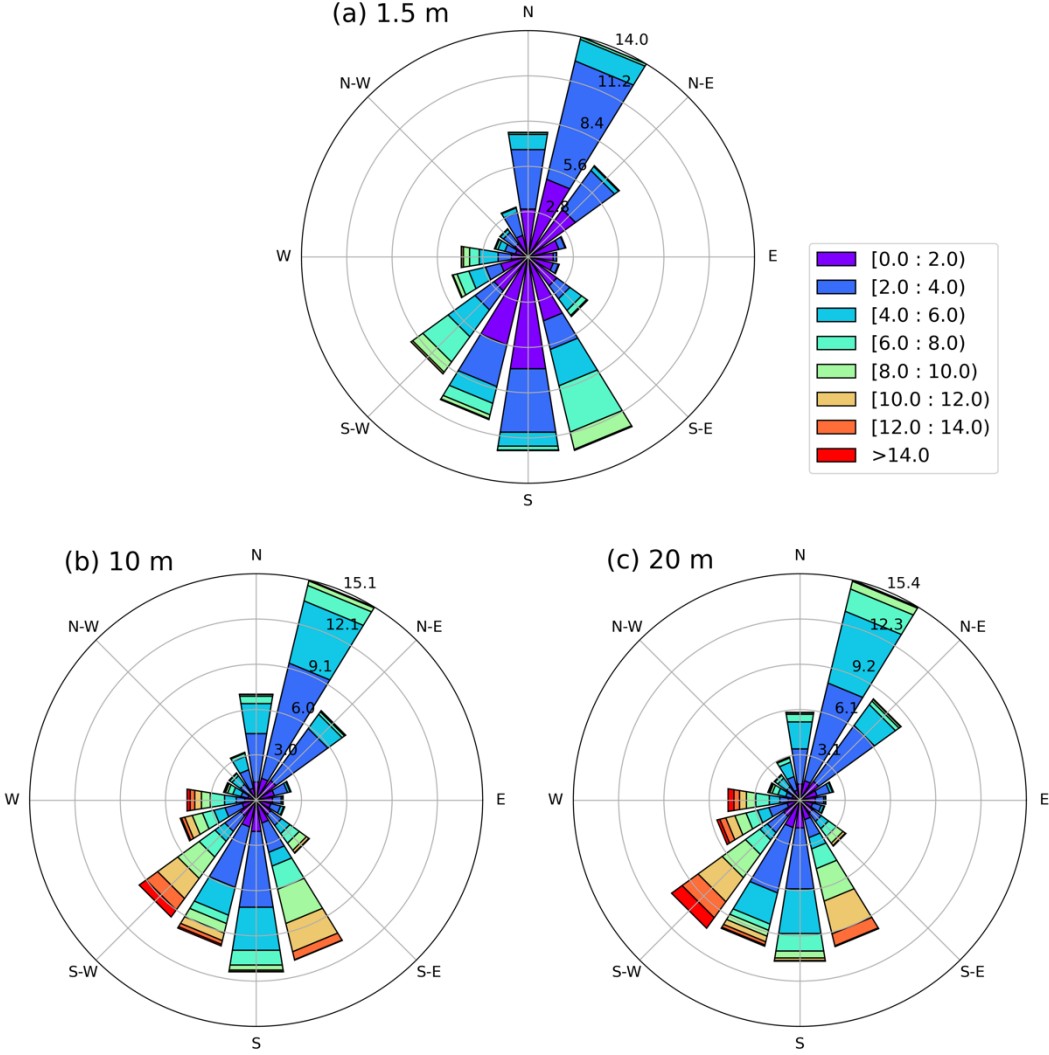

**Figure 2: Wind roses calculated from the 15-year 10-min wind data at (a) 1.5, (b) 10, and (c) 20 m at the QOMS station.**




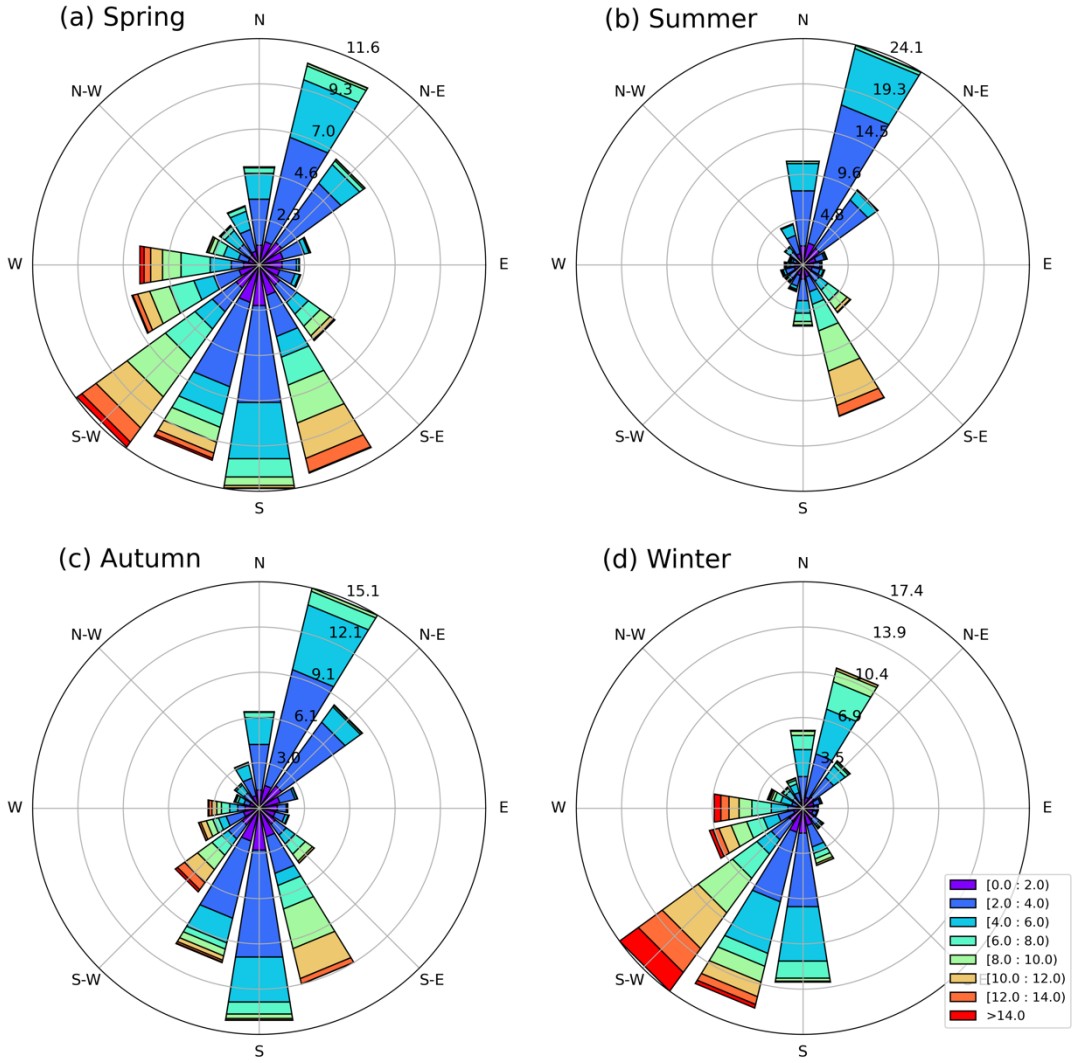

**Figure 3: Seasonal wind roses calculated from the 15-year 10-min wind data at 10 m height at the QOMS station. (a) Spring, (b) Summer, (c) Autumn, and (d) Winter.**

As indicated in Table 1, the continuity of the wind time series is the best at 10 m. Seasonal wind roses calculated from the

15-year 10-min wind time series at 10 m height are shown in Figure 3. Wind roses significantly differ in the four seasons in

terms of wind direction and speed. In spring, southerly wind (44.1%, from south-southeast to southwest) and north-northeast

wind (10.5%) dominate (Figure 3a). Northerly wind speeds are generally less than 6.0 m/s, while a significant number of

southerly wind speeds exceed 8.0 m/s. In summer (Figure 3b), the winds primarily blow from north-northeast (24.1%) and



south-southeast (16.4%) directions, and the south-southeast wind speed is significantly larger than the north-northeast wind
       speed. The percentage of wind from other directions is very low. In autumn (Figure 3c), southerly wind (36.8%, from south-
       southeast to south-southwest) and north-northeast wind (15.1%) dominate. In winter (Figure 3d), southwest wind (17.4%) is
       predominant, followed by south-southwest wind (15.5%), south wind (13.2%), and north-northeast wind (11.0%). The
       southwest wind speed is also significantly larger than the wind from other directions. In general, southerly wind and
northerly wind are prevailing wind directions at the QOMS station, and the southerly winds are stronger than the northerly
       winds. Winter and spring wind speeds are larger than in the summer and autumn seasons. Interestingly, there are almost no
       west winds in summer. However, the percentage of westerly winds increases, as well as the wind speed gradually increases
       from autumn to spring.

The monthly average wind speed at 10 m from September 2005 to June 2019 and the annual mean wind speed from 2006 to
       2018 are shown in Figure 4. The monthly average wind speed fluctuates mainly between 3.5 and 7.0 m/s and is characterized
       by pronounced seasonal variations, with high wind speed in winter and low wind speed in summer (Figure 4a). The annual
       mean wind speed fluctuates slightly around 4.6 m/s. The changing trend in mean wind speeds from 2006 to 2018 is not
       detectable, though it seems to exhibit a very slight decreasing tendency (Figure 4b). The wind speed trend at the QOMS
station differs from other findings, which have observed a significant increase in near-surface wind speeds across the TP and
       China since 2002 (Zhang and Wang, 2020; Zhang et al., 2024).

       Predominant wind directions at the QOMS station are slightly different in the four seasons, although the wind blows
       generally along the river valley. Moreover, wind speeds are significantly different in different directions. Due to the distinct
characteristics of wind in different seasons, we analyzed not only the full-scale spectra of wind but also the seasonal spectra
       in next sections.





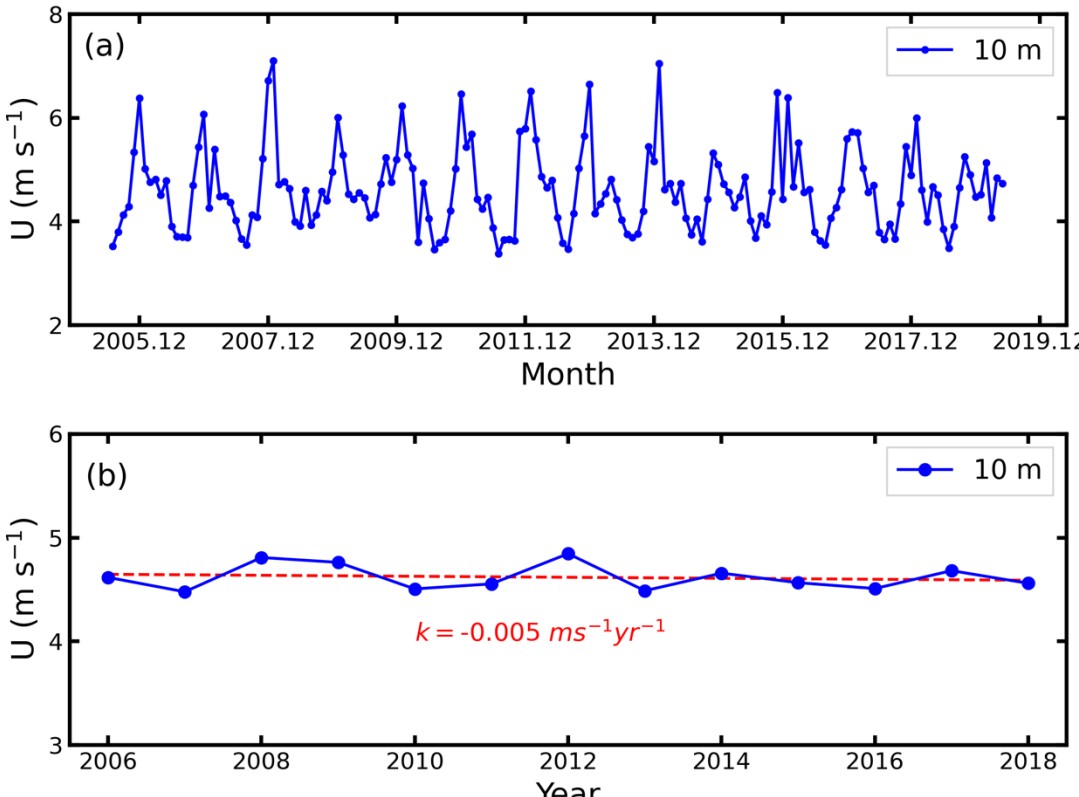

**Figure 4: Time series of monthly averaged wind speed at 10 m from September 2005 to June 2019 (a) and the annual wind speed from 2006 to 2018 (b). The dashed straight line indicates the linear trend of annual wind speed during 2006~2018, and k is the slope of the straight line.**


## 3.2 Full-scale spectra

As shown in Table 1, the data availability of the 10-min wind data from 1.5 and 20 m was below 90% in multiple years. Thus, 10-min wind data observed by the PBL tower at 2, 4, 10 m heights were used to calculate the wind speed spectra in the

frequency range from about 10 yr$^{-1}$ to 20 min$^{-1}$. As described above, the missing data points were interpolated using the linear interpolation method. Moreover, 10-Hz 3D wind data from 2015 and 2016, which have 100% data coverage on the daily time scale, were used to calculate daily wind speed spectra in the frequency range from about 1 day$^{-1}$ to 5 Hz. Figure 5 shows the full range frequency-weighted spectra $fS(f)$ of wind speed, with $f$ from about 10 yr$^{-1}$ to 5 Hz. Solid curves in Figure 5 are spectra calculated from the 15-year long time series of 10-min horizontal wind speed data, which shows a more



climatologically representative spectrum. The dotted and dashed curves are the composite daily wind spectra calculated by

using the daily 10-Hz 3D wind data observed in 2015 and 2016, respectively.

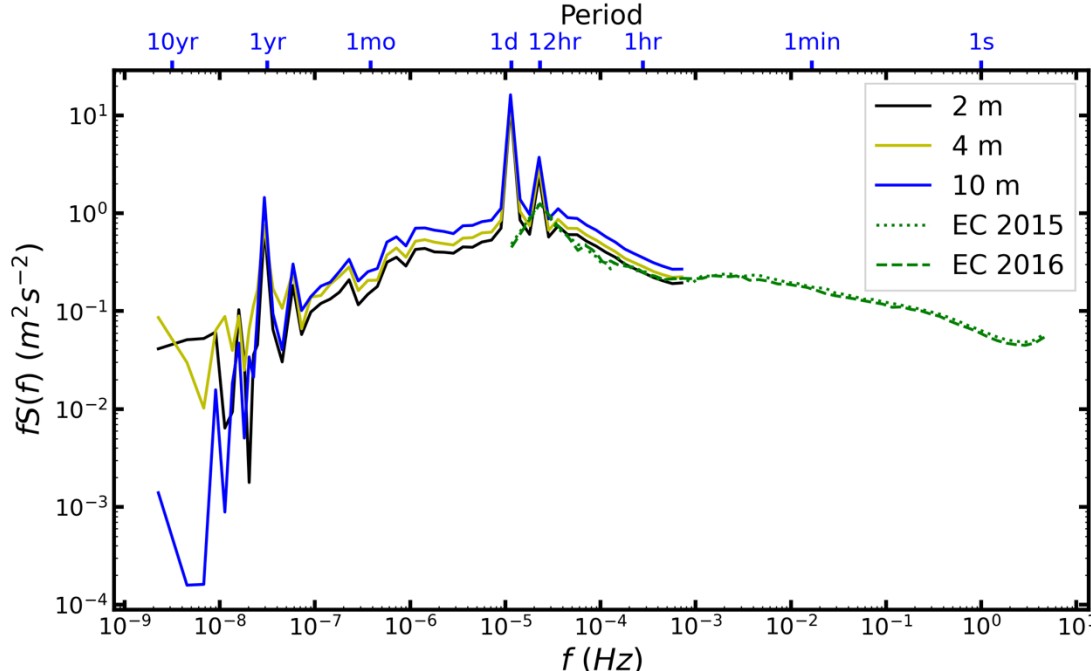

**Figure 5: The frequency-weighted spectra fS(f) as a function of frequency f of horizontal wind speed at 1.5, 4, 10, 20 m heights at the QOMS station calculated from the 15-year 10-min wind data (solid curves). The dashed curve is averaged daily spectra**
**calculated from the 10-Hz 3D wind data from year 2016.**

In the 15-year full-scale spectra, there are multiple spectra peaks clearly observed. The first narrow peak is at 1yr$^{-1}$. The

yearly peak has been frequently observed in the full-scale spectra at costal, offshore, and terrestrial sites (Kang and Won,

2016; Larsén et al., 2016; Watson, 2019). The second peak is the diurnal peak, which is associated with surface heat flux

modulations. The diurnal peak is often found to be missing or insignificant at an offshore or coastal site (Larsén et al., 2013;

Larsén et al., 2016), while it is the most significant peak at an inland site (Horvath et al., 2012; Kang and Won, 2015, 2016).

The diurnal peak at the QOMS station is even more significant than previous findings (Horvath et al., 2012; Kang and Won,

2016), the power density of the peak is larger than neighbor power densities by more than one order of magnitude.

Interestingly, there is a third peak at the frequency of 12 hr$^{-1}$, which is not frequently observed either at a costal/offshore site

nor at an inland site. The 12 hr$^{-1}$ peak is due to the daytime cycle between morning calm wind and afternoon strong southerly

wind, which is the result of interactions between the subtropical westerly jet and local valley winds at the QOMS site (Sun et

al., 2018).




The transition between spectra calculated from 10-min wind time series and from 10-Hz 3D wind data is rather smooth,
indicating the consistency of the two types of data time series in the overlapping frequency range. The spectra calculated
from 10-Hz 3D wind data are almost no difference between 2015 and 2016, which means one year ensemble mean is long
enough to have a good climatological representative of the daily spectra covering frequency range from 1 day$^{-1}$ to 5 Hz.
Moreover, similar to the 10-min spectra, there is also a peak at the frequency of 12 hr$^{-1}$ in the 10-Hz spectra.

### 3.3 Seasonal spectra

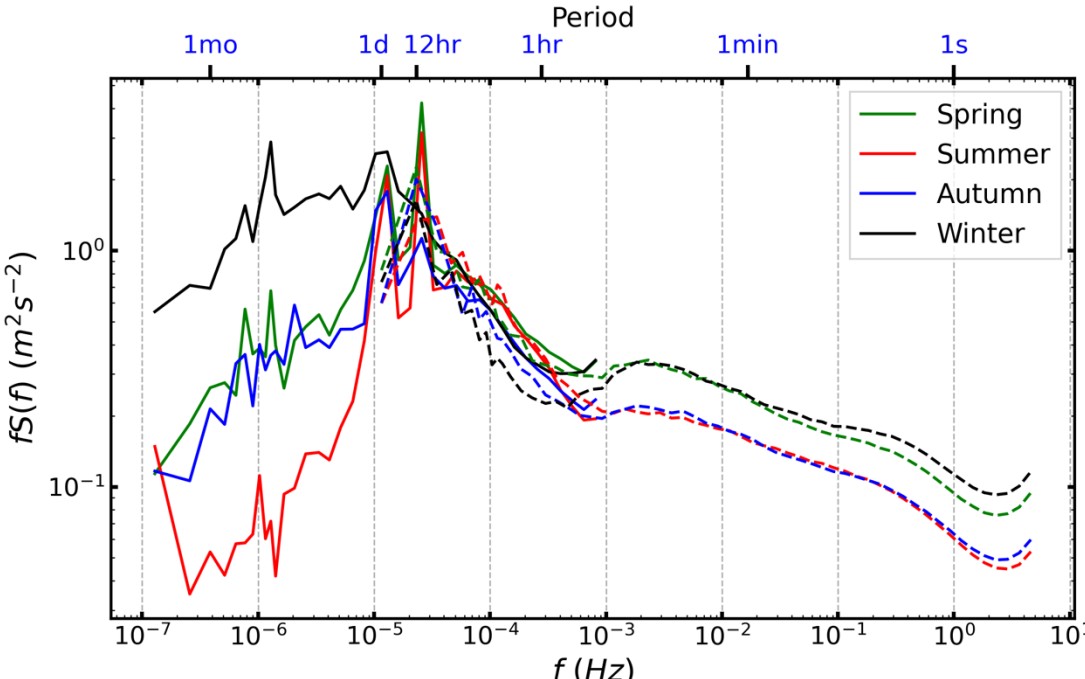


**Figure 6: The frequency-weighted seasonal spectra fS(f) as a function of frequency f of horizontal wind speed at 10m height at the
QOMS station calculated from the 15-year 10-min wind data (solid curves). The dashed curve is averaged daily spectra in four
different seasons calculated from the 10-Hz 3D wind data from year 2016.**

To reveal the seasonal characteristics of the spectra of the 15-year wind speed time series, we divide the year into four
seasons according to the following rules: March, April, and May for spring, June, July, and August for summer, September,
October, and November for autumn, and December, January, and February for winter. For each season of the year, we
calculated the spectrum of the 3-month wind speed time series. And then, for each season, we averaged the spectra at each $f$
over the 15 years to obtain a 15-year composite seasonal spectrum. As data continuity of the 10-min tower data is best at 10



m height, thus we only calculated seasonal wind spectra at 10 m height at the QOMS station. Moreover, 10-Hz 3D wind data from 2015 and 2016 were also used to calculate the seasonal spectra, and then the spectra were averaged over the two years. Finally, full-scale seasonal spectra were obtained with $f$ ranging from about 3 mo$^{-1}$ to 5 Hz (Figure 6).

As shown in Figure 6, the diurnal peak of the spectra in winter season is less sharp than in other seasons, which is also
reported by (Kang and Won, 2016). Moreover, the 12 hr$^{-1}$ peak somewhat disappears in winter, and the 12 hr$^{-1}$ peak in spring and summer seasons is more significant than the diurnal peak. On the low-frequency side of the diurnal peak ($f \geq 1$ day$^{-1}$), spectral density in winter is much higher than in other seasons. Spectral densities in spring and autumn are similar and are in between winter and summer. In winter, the QOMS station is mainly controlled by strong westerly winds, as indicated in Figure 3 and Figure 4. Strong wind and frequent synoptic weather events lead to the high spectral density in winter. The low
spectral density indicates synoptic weather events are significantly less frequent in summer than in other seasons. In the mesoscale frequency range (1 day$^{-1} \leq f \leq 1$ hr$^{-1}$), spectral densities in the four seasons have no significant differences. Interestingly, the summer spectral density is comparable to other seasons and even slightly larger than the winter spectral density in the high-frequency side of the mesoscale frequency range. This is likely due to the diurnal variation in wind speed and direction reported by Sun et al. (2018) and the frequent daytime convective activities in summer in the Mt. Everest
region. In the microscale frequency range ($f \geq 2 \times 10^{-3}$ Hz), spectral densities in winter and spring are significantly larger than in summer and autumn. Spectral density in winter is slightly larger than in spring, and spectral density in summer is very similar to autumn. Generally, near-surface turbulences are generated thermally and mechanically. Högström et al. (2002) reported that turbulence at frequencies higher than 10$^{-3}$ Hz is generated mainly by wind shear in the surface layer between the ground surface and the measurement height. As indicated in Figure 4a, wind speed in winter is significantly larger than
summer. Thus, the strong wind speed and wind shear explain the high spectral density in winter. Kang and Won (2016) investigated seasonal spectra using wind data observed at the Boulder atmospheric observatory facility in Erie, Colorado. Also, they found variations in the seasonal spectra in the microscale frequency range ($f \geq 2 \times 10^{-3}$ Hz), while with spring and summer spectral densities slightly larger than winter and autumn. The different behaviors in seasonal spectra between this study and Kang and Won (2016)'s results are related to the different climatic conditions, land surface characteristics, and
topography.

All frequency-weighted seasonal wind speed spectra ($fS(f)$) fall off as $f^{2/3}$ in the frequency range between $3 \times 10^{-5}$ and $6 \times 10^{-4}$ Hz, which seems to be in the Kolmogorov inertial subrange. The winter spectrum falls off even earlier at the frequency of $1 \times 10^{-5}$ Hz. At the frequency around $1 \times 10^{-3}$ Hz, there is a slope transition zone, and the spectral density increases beyond the
frequency of $1 \times 10^{-3}$ Hz but with different magnitudes in different seasons. Thus, the transition is the most pronounced in



winter, while is obscure and becomes a plateau in summer. In the microscale frequency range ($f \geq 2 \times 10^{-3}$ Hz), spectral densities fall off more smoothly.

### 3.4 Spectra gap

In the transition range between mesoscale and microscale, at frequencies around $1.0 \times 10^{-3}$ Hz, a spectral gap is frequently
observed (Van der Hoven, 1957; Fiedler and Panofsky, 1970; Högström et al., 2002; Larsén et al., 2016). Due to the poorly understanding of the interactions between 2D and 3D turbulences in this transition range, debate on the existence of the gap is still ongoing. In this study, obvious spectral gap around $1.0 \times 10^{-3}$ Hz exists in the 15-year full-scale spectra (Figure 5) and seasonal spectra (Figure 6).

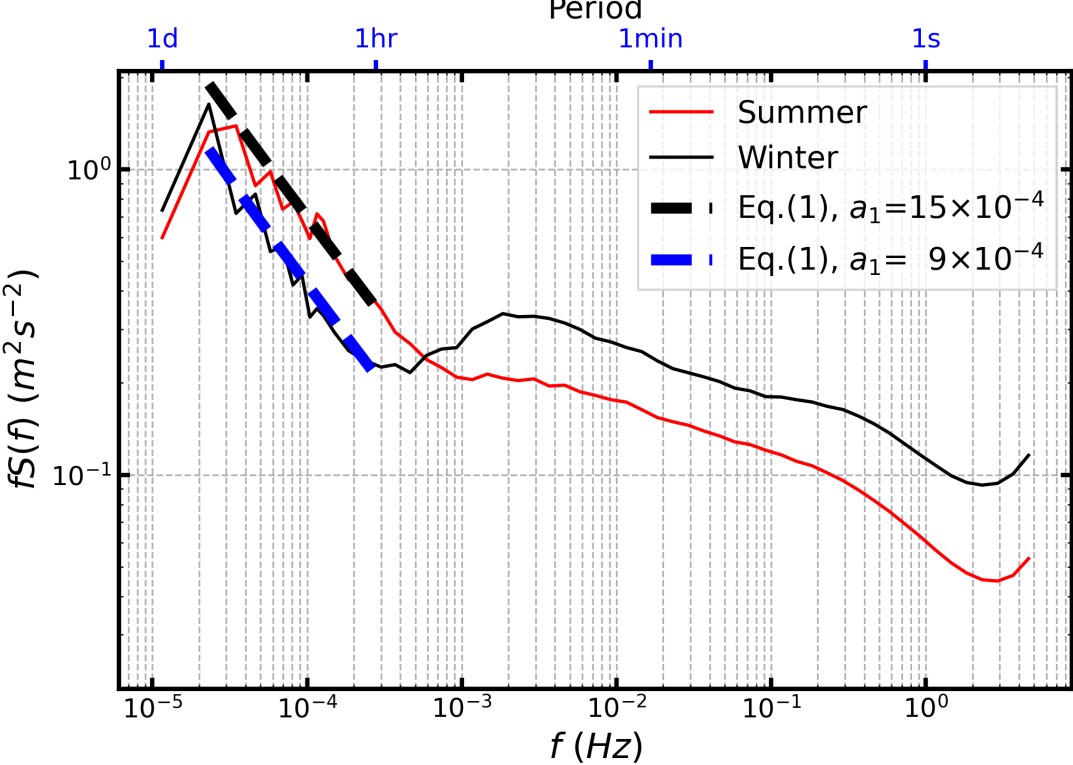

**Figure 7: The frequency-weighted summer and winter daily spectra fS(f) as a function of frequency f of horizontal wind speed calculated from the 10-Hz 3D wind data collected in 2015 and 2016. The thick black and blue lines indicate Equation (1) with different a1 values.**

As the wind speed, wind direction, and spectral characteristics significantly differ between summer and winter seasons, daily
spectra in these two seasons were selected to investigate the spectral gap in terms of its generality and behavior. The 10-Hz



3D wind data collected in 2015 and 2016 were used to the calculate the daily spectra, and only days without missing data points were selected. Figure 7 shows the frequency-weighted summer and winter daily spectra of horizontal wind speed. There is pronounced spectral density minima at $4.5 \times 10^{-4}$ Hz in winter spectrum, indicating the spectral gap which is also the transition from mesoscale 2D flow to 3D boundary-layer turbulence. The spectral gap around $1.0 \times 10^{-3}$ Hz is almost vanished in the summer spectrum. The spectral density in summer is higher than in winter for frequencies lower than the spectral gap, while above the gap, the spectral density in winter increases significantly and surpasses that of summer. This results in a more noticeable gap in winter than in summer. The frequency range higher than $1.0 \times 10^{-3}$ Hz represents the classical 3D boundary-layer turbulence region and the frequency range $5.0 \times 10^{-3} < f < 2.0 \times 10^{-1}$ Hz is known as the shear production range (Tchen et al., 1985; Högström et al., 2002; Larsén et al., 2016). Consequently, the turbulence intensity, especially the shear production, is more pronounced during winter than in summer at the QOMS site.

The spectral structure at frequencies lower than the spectral gap has been less studied and understood (Högström et al., 2002; Larsén et al., 2013; Larsén et al., 2016). Larsén et al. (2016) proposed a model to express the spectral structure in the mesoscale range that the weighted spectrum $fS(f)$ decreases with frequency with a $f^2$ slope and followed by a $f^{2/3}$ slope as

$$S(f) = a_1 f^{-2/3} + a_2 f^{-2}, \tag{1}$$

with $a_1 = 3 \times 10^{-4}$ m$^2$s$^{-8/3}$ and $a_2 = 3 \times 10^{-11}$ m$^2$s$^{-4}$. Horizontal wind speed spectra observed at coastal onshore and offshore sites have been proved to match the spectral model perfectly (Larsén et al., 2013; Larsén et al., 2016). As to whether the generality of Equation (1) applies to the spectra of mountainous regions, they considered that further examinations were needed. Equation (1) is plotted as the dashed blue line in Figure 7 but with $a_1 = 9 \times 10^{-4}$ m$^2$s$^{-8/3}$ and $a_2 = 3 \times 10^{-11}$ m$^2$s$^{-4}$, which describes the winter spectra in the mesoscale frequency range very well. For the summer spectra, $a_1 = 15 \times 10^{-4}$ m$^2$s$^{-8/3}$ needs to be used and $a_2$ remains the same. Thus, Equation (1) is applicable in mountainous regions to describe horizontal wind spectra in the mesoscale frequency range. Note that the second term on the right-hand side of Equation (1) becomes less important as the frequency increases and the spectra simply converge to $f^{2/3}$ (figures are omitted). Similar results were reported by Kang and Won (2016).



**Figure 8: Observed horizontal wind velocity spectrum (the solid black curve), the spectral model Equation (1) extended to high frequencies (the dashed green curve), the Kaimal spectrum for frequencies lower than the peak frequency (the dotted blue curve), and the merged spectrum of spectral model equation (1) and Kaimal spectrum (the dashed red curve) for (a) summer and (b) winter seasons.**





Larsén et al. (2016) concluded that the gap region of the spectra of horizontal wind observed from coastal and offshore sites can be modelled under the assumption that the 3D turbulence and the 2D mesoscale variations are uncorrelated. In this study, in line with Larsén et al. (2016)'s hypothesis, microscale and mesoscale wind spectra are linearly composited. The Kaimal spectrum (Kaimal and Finnigan, 1994) is used for microscale range for frequencies lower than the spectral peak ($f_p$), while

Equation (1) is used for the mesoscale range. The Kaimal spectrum for horizontal wind velocity is expressed as

$$\frac{f S_V(f)}{u_*^2 \phi_\epsilon^{2/3}} = A\left(\frac{fz}{\overline{V}}\right)^{-2/3}, \tag{2}$$

$$\phi_\epsilon^{2/3} = \begin{cases} 1 + 0.5|z/L|^{2/3}, & z/L \leq 0 \\ (1 + 5\,z/L)^{2/3}, & z/L \geq 0 \end{cases}, \tag{3}$$

where $S_V(f)$ is the spectrum of horizontal wind velocity, $u_*$ is the friction velocity, $\phi_\epsilon$ is the dissipation rate of turbulent kinetic energy, $z$ is the observation height above ground, $\overline{V}$ is the averaged horizontal wind velocity, $L$ is the the Obukhov length, $A$ is a constant and is taken as 0.35 in this study. Figure 8 shows the superimposed spectral components. It is shown

that the linear superimposition of the Kaimal spectrum (dashed red curve) and mesoscale spectral model Equation (1) matches the observed spectrum (solid black curve) very well. Thus, the hypothesis also holds for the QOMS site, which is surrounded by complex topography. This theory also accounts for the more noticeable spectral gap in winter compared to summer. During winter, the 3D turbulent spectral energy is more pronounced, while the contribution of 2D mesoscale eddies

to spectral energy is less, resulting in a more visible spectral gap in winter than in summer.

## 4 Summary and conclusions

15-year time series of horizontal wind speed observed at a mountainous site on the north slope of Mt. Everest were used to investigated the full-scale spectra of horizontal wind speed. The Characteristic of wind speed and wind speed spectrum were analyzed. The findings of this study advance the understanding of the wind speed spectrum in mountainous regions.


The prevailing wind directions at the QOMS station are from the south and north. Wind speed shows significant seasonal variation, with stronger speeds in winter and weaker speeds in summer. The annual average wind speed shows almost no detectable trend from 2006 to 2018, although exhibiting a very slight decreasing tendency. The changing trend in wind speed at the QOMS station differs from findings in other regions, which have observed a significant increase in near-surface wind

speeds in the TP and China since 2002.

In the 15-year full-scale spectra, we identified three peaks at frequencies around 1 yr[-1], 1 day[-1] and 12 hr[-1], respectively. The 12 hr[-1] peak is significant in spring and summer while it disappears in winter. The 1 day[-1] peak is usually insignificant offshore and coastal sites but is frequently observed at terrestrial sites. However, the wind spectral peak at the frequency of

12 hr$^{-1}$ is only observed at some mountainous or urban sites. The spectral peak at 12 hr$^{-1}$ is most likely related to the unique daytime local circulations in the valley where the QOMS station is located. The seasonal spectra show distinct characteristics in different seasons. On the low-frequency side of the diurnal peak ($f \leq 1$ day$^{-1}$), the spectral density is the highest in winter, which is associated to the strong westerly wind and frequent synoptic weather events during the winter season. On the high-frequency side of the spectral gap, the spectra density in winter and spring is much higher than in summer and autumn,

suggesting that at the QOMS station, the turbulence intensity, especially that generated by shear, is more significant in winter than in summer. This is due to the strong winds during the winter and spring seasons.

The spectral gap is observed around the frequency of 10$^{-3}$ Hz. It is the most obvious in the winter season while is almost vanished in the summer. The spectral intensity in the mesoscale range in winter is lower than in summer while the spectral

intensity in the microscale range in winter is higher than in summer, leading to the most pronounced spectral gap in winter. Larsén et al. (2016) hypothesized that the 3D microscale turbulence and 2D mesoscale variations are uncorrelated and the microscale and mesoscale wind spectra are linearly composited. Under this hypothesis, the gap region of the horizontal wind spectrum is modelled very well at the mountainous QOMS site.

**Acknowledgments**

This project has received funding from the National Key R&D Program of China (grant no. 2022YFB4202104) and the National Natural Science Foundation of China (grant nos. 42230610, 41975013). We gratefully acknowledge the colleagues at the QOMS site for their diligent maintenance of the instruments.

**Code/Data availability**

All the data used in this study have been published in Ma et al. (2020) and are publicly accessible at the National Tibetan

Plateau Data Center (https://doi.org/10.11888/Meteoro.tpdc.270910, last access on June 12, 2024). The codes for data processing are available upon request.

**Author contribution**

Conceptualization: Cunbo Han, Yaoming Ma, Weiqiang Ma.
Methodology: Cunbo Han, Fanglin Sun, Yunshuai Zhang.

Visualization: Cunbo Han.
Funding acquisition: Cunbo Han, Yaoming Ma, Fanglin Sun.





Writing (original draft): Cunbo Han.

Writing (review and editing): Yaoming Ma, Weiqiang Ma, Fanglin Sun, Yunshuai Zhang, Hanying Xu, Wei Hu, Chunhui Duan, Zhenhua Xi.

**Competing interests**

The authors declare that they do not have any competing interests.

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
