# Peer review of "Full-scale spectra of 15-year time series of near-surface horizontal wind speed on the north slope of Mt. Everest"

_EGUsphere, 2024_

## Referee Comment (RC2)

**Review of:  Full scale spectra of 15-year time series of near-surface horizontal wind speed on the north slope of Mount Everest, by Han et al.**

**General Comments**

This paper presents and discusses long-term windspeed spectra based on data collected in a mountain valley 30 km north of Mt Everest – a result that is unique and interesting because of the location, but otherwise not representing little that is new.  Moreover, there are some inaccurate statements and in interpretation and presentation, as described below.  Thus, significant effort to make this paper suitable for publication should the results be deemed sufficiently novel.

1.   The authors use the assumption that the high- and low-frequency portions of the spectrum are uncorrelated (quoting Larsén et al. 2016) to conduct a curve-fitting exercise that reproduces part of the observed spectrum using the characteristic 2-d slope for lower frequences and invoking a formula for Kaimal and Finnigan at higher frequencies, presumably weighting it to produce the right result.

    Why is this flawed?  First, it isn't obvious that a successful curve fitting is associated with lack of correlation.  In a classic paper, Charnock (1957), he showed that a spectral gap had to be sufficiently large for terms containing products of high-and-low frequency terms to average to zero – i.e., for the average of the nonlinear terms to be zero.  For the product of two variables – e.g.., vertical velocity and horizontal zonal wind, to be zero, the spectral gap has to be between $f_c/2$ and $3f_c/2$, where  is the cutoff frequency for a filter separating the high- and low-frequency motions.  Using Charnock's approach for products of three terms, the spectral gap has to be larger, i.e., between $f_c/3$  and $5f_c/3$ (LeMone et al. 1976, p 1315).  Thus, the idea of no correlation – i.e., no interaction between the synoptic (2-D) and turbulence (3-D) motions is not necessarily valid even with a gap.  In fact, Figure 5 shows that stronger synoptic-scale motions in winter are associated with stronger turbulent motions, -- a correlation, since the stronger turbulence results from the stronger winds driven by synoptic features.

2.  Equation (1) doesn't really replicate the actual spectra with the coefficients being constant (except for fitting the regions with -2 and -2/3 slopes – which has to be the case, correlation or no correlation. The need for a varying function is demonstrated through use of the Kaimal equations (2) and (3).

3.  The 12-hr peak in the windspeed data is to be expected for any mountain location, and not unique to the QOMS station.  One need merely to refer to Chapter 11 of David Whiteman's *Mountain Meteorology.*  (references at the end of this review). Likewise, atmospheric tides produce a 12-hr cycle in the surface winds over at least the tropical ocean.

*Mountain Locations:*  There are several examples of mountain-valley, mountain-plains, etc. circulations that go upslope during the day, and downslope during the night.  This is a 24-hour cycle in the up-and-downslope component of the wind, but a 12-hour cycle in the wind speed.  (This is a classic example of the misleading nature of computing spectra of windspeed instead of dividing the wind into components). Moreover, the authors seem to emphasize only the daytime wind with the 12-hour peak (Line 230). Actually, if there were only a wind peak during the day with no wind change at night, that should show up mostly as a 24-h peak rather than a 12-hr peak.  (I see that the observations in Sun et al. (2018) show a nighttime downslope wind peak as well as the daytime peak).

*Atmospheric Tides.*  The authors mention (L23 in the abstract and elsewhere) that a 12-hour peak is rarely observed over the ocean.  This is untrue.   In fair weather over the tropical oceans, the sea-surface temperature doesn't change that much, muting the normal 24-h cycle. Thus, in fair weather, atmospheric tides can be revealed in the surface wind.  I have four references, the earlier ones indicating that semidiurnal surface-wind variation was known for over 50 years. The first two are based on summer data from multiple ships in the 1974 GARP Atlantic Tropical Experiment, GATE, which took place over the tropical Atlantic – unfortunately in hard-to-access papers.  The third is from the classic book on atmospheric tides by Chapman and Lindzen.  The 4[th], a more recent and accessible publication, provides a lot of useful information.   While the data differ, all show atmospheric tides show up in the surface wind over the ocean in the same amplitude range.

Jacobs (1980, figure 3) used "all GATE data" (i.e., all ship data) to produce a composite diurnal plot showing semidiurnal wind *speed* peaks at an average maximum of 4.4 m/s and average minimum of 0.5*(4.2 + 3.9) = 4.05, yielding a range of 0.35 m/s and amplitude of 0.175 m/s.

LeMone (1980, Figure 17)) showed diurnal-average curves, based on seven ships (6 from booms, 1 buoy).  The10-m zonal wind  has a semidiurnal pattern (2 maxima averaging 2.24 m/s,2 minima averaging1.54 m/s) with a range of 0.7 m/s and an amplitude of 0.35 m/s for 30 August to 20 September.  The meridional  wind has no semidiurnal cycle.

Chapman and Lindzen (1970, table 2S.9 shows the influence of atmospheric tides on surface winds (amplitude of () = (0.08,0.21) m/s) using data from an island site (Terceira, Azores) and combines data for the whole year.

Ueyama and Deser (2008) annual average amplitude, TAO buoys, $(u, v)$ = (0.14,0.06) m/s (Fig. 4); June-Nov amplitude: $(u, v)$ = (0.15,0.07) (Fig. 7)  (estimates rough). The Paper also has useful references.  This paper also shows a diurnal cycle, comparably small.

4. The introduction is sloppy, mixing references of observations from fixed points (like the observations here, and those from aircraft, and mixing spectra of wind speed and spectra of the horizontal wind components, and not necessarily covering the range of frequencies of interest.

   For example, the Sun and Lenschow paper cited uses aircraft data, which weren't necessarily sampled parallel to the wind. Second, they compute spectra of the wind components rather than speed. Third, their frequency range is severely limited by the length of the flight tracks and thus does not sample the synoptic (2-D) scale. Rather, their lower frequencies, in TOGA COARE at least, are likely associated with precipitating convection. Thus, their results have little to do with the theme of this paper. Focusing only on the papers dealing with the gap between synoptic and turbulent motions would greatly shorten and improve the introduction.

   Since the current introduction deals with spectral gaps that involve only boundary-layer motions, it should be noted that the presence or absence of a gap is a function of the sampling strategy.

   That is, if you are sampling from a point (or fixed tower), you will likely get a two-peak spectrum in the convective boundary layer in the presence of horizontal convective rolls (which are nearly parallel to the wind and take of the order of 30 min to an hour to pass a point on the surface) that coexist with large 3-d convective eddies (which pass a point in more like 5-10 min). This separation in periods is useful for separating out the two structures for analysis, even though the structures of roughly the same scale in the crosswind direction. With only along-wind data, one might think that the spectral gap is big enough for zero nonlinear terms under Charnock's criterion.

   However, if you sample using an aircraft flying normal to the roll axis, you get a single spectral peak, corresponding to the roll/convective cell combination. As can be seen in clear-air radar echoes in the presence of rolls, the convective cells (large eddies) lie in the rolls' upwelling portion – i.e., they are tightly correlated. So, even if you get the spectral gap identified by Charnock as suggesting zero nonlinear terms with the along-wind data, the crosswind data tell another story.

   Thus, unlike what is stated on line 72, a spectral gap does NOT mean that motions are weakly correlated. In fact, they are strongly correlated and interact in this case. Clearly, the presence of the convective cells in the upwelling portion of the rolls indicates a correlation. A paper you cite, LeMone (1976) discusses the interaction of the rolls and cells.

5. Moreover, I don't think that a list of spectral peaks at different geographic locations is illuminating unless they are carefully grouped according to terrain and sampling strategy (length, seasons, fixed vs traveling platform and whether/how its track is related to wind direction).

**Specific comments:**

L65. Lenschow and Sun applies to aircraft data, not data collected along the wind. Also, they show spectra of the wind components, which is different from what is done here. Should check these studies to see if they correspond to what is done in this paper.

In the introduction, the discussion is longer than it needs to be. What is of interest here is the gap between synoptic (essentially 2-D) and turbulence (3-D) motions, rather than a gap between structures in the boundary layer.

L82-84. Height dependence depends on the variable observed. With rolls as an example, the spectral gap in vertical velocity would become MORE evident with height. Of course, the horizontal winds vary more near the surface. (don't understand why you need a reference for this – it's common sense).

L90. Studying a map of Beijing, a 12-hour peak in wind speed seems likely, associated with the 24-hour upslope-downslope wind cycle with nearby higher terrain.

Figure 1. A terrain map would be more helpful. It appears that the site isn't on a slope but is in a valley. Details are important – see David Whiteman's *Mountain Meteorology* (reference below). Also the valley location would make the comparison with the wind rose more meaningful.

L118. 30 km north of the peak of Mt. Everest in a valley is a significantly better description than "on the north slope of Mt. Everest."

L160-165. A reasonable summary of the wind roses would start with a statement about how much the winds are along the valley axis – I was surprised that this comparison isn't made until farther down in the paper and that it was not mentioned in the conclusions.

L193. Delete "changing."

L195-6. Are the increasing near-surface wind speeds over flatter terrain?

L225. This is hardly surprising and well-known that the diurnal cycle is stronger over land. Are references even needed? (i.e., this is discussed in most introductory meteorology books).

L230. Don't Sun et al. see the classic mountain-valley wind, with the daytime wind influenced by the westerly jet and a second night-time maximum associated with downslope wind? You have to consider the night-time wind as well. If there was only a

maximum in the 12-hour period during the day influenced by the subtropical westerly jet, that would be a 24-hour cycle. (see general comments)

L236-8. Not necessarily. Perhaps they would be close because the average winds were close.

L253. As expected, the mountain-valley system is stronger in the summer, and it is not surprising that a 12-h peak is stronger than the 24-h peak in wind SPEED. For the wind component along the slope, the period would be 24 h. This is an example of the misleading aspect of using speed instead of wind components.

L270. Regarding Kang and Won, what was the length of the dataset used?

A cautionary note: The relationship between scale in m and frequency in Hz is often a function of the windspeed (if features are carried by the wind). For winter, and stronger winds, the same scale of eddy (thinking about km-scale) will produce a higher frequency. For winter, that eddy might be smaller since the boundary layer could be smaller than in the summer.

Eq (1) doesn't make much sense unless the coefficents are functions. Without functions, how can it match any observations "perfectly" (L312).

Figure 8. and Eq (1). The line labeled "Eq. (1)" is only the low-frequency part, along with Kaimal and Finnigan's spectral function at the higher frequencies. This does not show that Eq. (1) works if the coefficients are constants. And it is curious that you don't extend Kaimal's spectral power to the higher frequencies.

L328. It does appear that you can linearly composite the low- and high-frequency data, but this does not imply that the two are independent of one another.

Figures 5-8. The rising spectral density at the high frequency end reflects the presence of white noise and should be eliminated or at least pointed out.

Figure 8. Did you evaluate periods for stable (L>0) and unstable (L<0) separately in Eq. (2)-(3) and then combine? There should be a more complete explanation about how the calculation is done. Also, more precise to say 'added' rather than 'merged."

6. Summary and conclusions:

L342. Again, misleading to write "on the north slope of Mt. Everest," since the location of the station in the valley has strong control over the wind direction.

L346. Along the valley! (and not north-south)

L353-4.  Not a result of this paper.

L354-5.  More accurate to say that the 12-h peak is common in spectra of wind SPEED.  This is an important distinction.  "Some urban sites" -- are these urban sites close to/in mountains, like Hong Kong or Beijing?  Again, not a result of this paper.

L356.  The 12-h period is not unique; nor is it related to the single peak during the day, as explained earlier.

L363.  That's only 16.7 minutes, well within convective boundary layer frequencies!  If you look at the curves carefully, the minima tend to be at lower frequencies, which makes more sense.  You have a better estimate in the body of the paper. (L298)

L366-367.  For the many reasons (and based on the results) the synoptic and turbulence spectra ARE correlated.

REFERENCES

Chapman, S., and R. S. Lindzen, 1969:  Atmospheric Tides.  D. Reidel Publishing Company, the Netherlands, 200 pp.

Jacobs, C.A., 1980: Mean diurnal and shorter period variations in the air-sea fluxes and related parameters during GATE, In Siedler, Gerold, and John D. Woods, 1980:  Oceanography and Surface Layer Meteorology of the B/C-Scale, GATE, V1, Pergamon press, 294 pp. Supplement 1 to Deep-Sea Research Part A, v. 26, pp 65-98. https://doi.org/10.1016/B978-1-4832-8366-1.50009-1.

LeMone, M. A., 1980:  The marine boundary layer, pp. 183-231, in *Workshop on the Planetary Boundary Layer*.  J.C. Wyngaard, ed.  American Meteorological Society, 322 pp.

Ueyama, R. and Deser, C., 2008. A climatology of diurnal and semidiurnal surface wind variations over the tropical Pacific Ocean based on the tropical atmosphere ocean moored buoy array. *Journal of climate*, *21*(4), pp.593-607.

Whiteman, C. D., 2000:  *Mountain Meteorology:  Fundamentals and Applications*.  Oxford University Press.355 pp.

---

## Author Comment (AC1)

Response to Referee's comments:

We would like to thank the Editor and the Referee for the time and efforts handling the reviewing our manuscript. The constructive comments and suggestions were very helpful to improve the manuscript.

The Referee's original comments are formatted in black, while our point-by-point responses are formatted in **blue** font. All the corresponding revisions in the revised manuscript are indicated using the "Track Changes" function.

**Reviewer #1**

Comments on **Full-scale spectra of 15-year time series of near-surface horizontal wind speed on the north slope of Mt. Everest**

In this article, a 15-year wind time series of near-surface horizontal winds from the National Observation and Research station called QOMS is analyzed. The research station is located on the north slope of Mt. Everest. The authors have also examined horizontal winds' spectral characteristics during different seasons. The wind data comprises 10-min data from 2005 to 2019 at four heights: 1.5, 2, 4, 10, and 20m. High-frequency 10 Hz data from 2015 and 2016 are also studied for information about microscale turbulence.

Overall, the study is quite interesting and provides many insights about wind climatology in the area. I believe the study can be further improved by addressing the following queries:

We would like to thank the review for the helpful comments and suggestions for recognizing the contributions made by this work. Our point-by-point responses are found below.

1. The wind roses in Figures 2 and 3 show higher wind speeds from the south, but no explanation is given for this phenomenon. Do katabatic winds play a role in this speed-up as the cold, dense air flows down from the top and becomes less dense as it heats up in the valley, Especially during the winter season? Please comment on this.

   Thanks for your helpful comments and suggestions. You are right, katabatic winds play an important role in the strong south wind. Many studies have already investigated this topic. The reason is twofold, local-scale katabatic winds (or glacier winds) and large-scale circulations. Following your suggestions, we discussed the reason at the end of second paragraph of section 3.1. Please see lines 208 to 214 in the revised manuscript: "……*Many studies have reported the*

*phenomenon that the south wind is stronger than the north wind (Cai et al., 2007; Song et al., 2007; Sun et al., 2007; Sun et al., 2017; Sun et al., 2018). Recently, Sun et al. (2018) revealed that the strong south winds in non-monsoon season are dominated by downward momentum transport for westerly winds aloft, while during the monsoon season, the strong winds are driven by up-valley winds from the Arun Valley east of Mount Everest migrating into the Rongbuk Valley where the QOMS station is located. Moreover, katabatic winds driven by the along-valley temperature gradient between cold temperatures to the south over glacier surfaces and warm temperatures to the north can accelerate the wind speed…...”*

2. In Section 2.3, it is mentioned that "linear detrending is applied to the wind speed data time series". Sometimes, this can significantly reduce the low-frequency part of the power spectrum. It would be interesting to know whether linear detrending the time series affects the results in this article considerably or not.

We agree that applying linear detrending to wind velocity time series might influence the power spectrum at low frequencies. In Figure 1, we plotted out the frequency-weighted spectra calculated using the 15-year 10-min wind velocity data observed at the height of 10 m. The solid black curve is calculated from the original wind time series without using linear detrending, while the dashed blue one is calculated from the wind velocity data after applying linear detrending. As the reviewer pointed out, differences are observed at the low-frequency part, but only at the first multiple frequencies. Thus, we conclude that applying linear detrending to wind time series would not affect the results in this study.

[Figure]

Figure 1: The frequency-weighted spectra $fS(f)$ as a function of frequency $f$ of horizontal wind speed at the height of 10 m at the QOMS station calculated from

the 15-year 10-min wind velocity data. The solid black curve is the spectrum calculated from the original wind velocity data, while the dashed blue curve is the spectrum calculated from the wind velocity data after applying linear detrending.

3. The article focuses on horizontal winds, but it would be interesting to also look at the vertical wind spectrum obtained from sonic anemometer. This would explain the flow circulation in the valley and high-altitude mountains in the south. Plus, you could see the distinction between microscale 3D turbulence and mesoscale 2D turbulence on the frequency scale.

Thanks very much for your suggestions. Although we focused on the horizontal wind spectrum, we plotted also the vertical wind spectra for winter and summer calculated from the 3D sonic wind data in Figure 6 in the revised manuscript. We discussed the characteristics of vertical wind spectra at the end of last paragraph in section 3.3. Please see lines 315 to 319 in the revised manuscript: "……*Moreover, daily vertical wind spectra calculated from the 10-Hz sonic wind data for summer and winter are also plotted in Figure 6. Rather than decreases in horizontal wind spectra in the frequency range from approximately $2\times10-3$ Hz to $3\times10-1$ Hz, the vertical wind spectra increase monotonically. At higher frequencies ($f \geq 1\times100$ Hz), the shape of the vertical wind spectra is similar to the horizontal wind spectra, and the spectral densities are very close as well……*"

4. In Figures 7 and 8, while the low-frequency part of the spectra follows the $f^{-2/3}$ scaling, the high-frequency part does not follow the same scaling. What could be the reason behind this? According to studies such as Larsén et al., 2016 and Kaimal and Finnigan 1994, you should observe the same scaling in the high-frequency part. Similarly, the Kaimal spectrum (blue dots) can be extended for frequencies higher than $4\times10^{-3}$ Hz.

The $f^{-2/3}$ scaling refers to the inertial subrange, where energy is neither produced nor dissipated, but cascaded down to smaller scales. Indeed, in this study, we also observed the $f^{-2/3}$ scaling in the high-frequency part (See the figure below, which is the same as Figure 7 in the manuscript but with the reference $f^{-2/3}$ spectrum). Note that there was a smoothing issue in the daily spectra and we fixed the issue and have updated Figures 5~8 in the revised manuscript. The inertial subrange in this study is approximately in the frequency range from $2.0\times10^{-1}$ to $2.0\times10^{0}$ Hz, which is relatively narrow and is slightly different from the ones reported in Larsén et al., (2016) (in a wider frequency range from $3.0\times10^{-2}$ to $1.0\times10^{0}$ Hz). We believe the differences are from the complex topography and complicated local circulations around the QOMS station. Moreover, the daily spectra in this study are highly correlated the four spectral regimes proposed by Högström et al. (2002) in Figure 4 in his paper.

Regarding the Kaimal spectrum, we agree that is can be extended for higher frequencies. Kaimal spectra were derived from the Kansas experiment (Kaimal et

al., 1972), where is relatively flat and homogeneous. In this study, Kaimal spectrum is not extendible to higher frequencies. We speculate that the main reason is the highly complex topography and underlying surface characteristics around the QOMS site in the Mt. Everest region. Kaimal and Finnigan (1994) also reported that when flow passes over varying terrain and hills, the velocity spectrum in the high-frequency range changes. This is due to the greater contribution to kinetic energy from mechanically (shear) produced turbulence than we expected in a classical convective mixed layer, which leads to the increase in spectral density in the high-frequency range. The wind spectra over complex terrain and heterogeneous surfaces in the Mt. Everest region is a very interesting topic. In this study, we focus on the spectral gap in the transition range between mesoscale and microscale. In the future, we will further study the applicability of the Kaimal spectrum in the microscale spectral range in this region.

[Figure]

Figure 2: The frequency-weighted summer and winter daily spectra $fS(f)$ as a function of frequency $f$ of horizontal wind speed calculated from the 10-Hz 3D wind data collected in 2015 and 2016. The thick black line indicates the reference $f^{2/3}$ spectrum.

**References:**

Högström, U., Hunt, J. C. R., and Smedman, A.-S.: Theory and measurements for turbulence spectra and variances in the atmospheric neutral surface layer, Boundary-Layer Meteorology, 103, 101-124, https://doi.org/10.1023/A:1014579828712, 2002.

Kaimal, J. C., Wyngaard, J. C., Izumi, Y., and Coté, O. R.: Spectral characteristics of surface-layer

turbulence, Quarterly Journal of the Royal Meteorological Society, 98, 563–589, https://doi.org/10.1002/qj.49709841707, 1972.

Kaimal, J. C. and Finnigan, J. J.: Atmospheric boundary layer flows: Their structure and measurement, Oxford University Press, New York1994.

---

## Author Comment (AC2)

Response to Referee's comments:

We would like to thank the Editor and the Referee for the time and efforts handling the reviewing our manuscript. The constructive comments and suggestions were very helpful to improve the manuscript.

The Referee's original comments are formatted in black, while our point-by-point responses are formatted in **blue** font. All the corresponding revisions in the revised manuscript are indicated using the "Track Changes" function.

**Reviewer #2**

General Comments

This paper presents and discusses long-term windspeed spectra based on data collected in a mountain valley 30 km north of Mt Everest – a result that is unique and interesting because of the location, but otherwise not representing little that is new. Moreover, there are some inaccurate statements and in interpretation and presentation, as described below. Thus, significant effort to make this paper suitable for publication should the results be deemed sufficiently novel.

1.  The authors use the assumption that the high- and low-frequency portions of the spectrum are uncorrelated (quoting Larsén et al. 2016) to conduct a curve-fitting exercise that reproduces part of the observed spectrum using the characteristic 2-d slope for lower frequences and invoking a formula for Kaimal and Finnigan at higher frequences, presumably weighting it to produce the right result.

    Why is this flawed? First, it isn't obvious that a successful curve fitting is associated with lack of correlation. In a classic paper, Charnock (1957), he showed that a spectral gap had to be sufficiently large for terms containing products of high-and-low frequency terms to average to zero – i.e., for the average of the nonlinear terms to be zero. For the product of two variables – e.g.., vertical velocity and horizontal zonal wind, to be zero, the spectral gap has to be between $fc/2$ and $3fc/2$, where is the cutoff frequency for a filter separating the high- and low-frequency motions. Using Charnock's approach for products of three terms, the spectral gap has to be larger, i.e., between $fc/3$ and $5fc/3$ (LeMone et al. 1976, p 1315). Thus, the idea of no correlation – i.e., no interaction between the synoptic (2-D) and turbulence (3-D) motions is not necessarily valid even with a gap. In fact, Figure 5 shows that stronger synoptic-scale motions in winter are associated with stronger turbulent motions, -- a correlation, since the stronger turbulence results from the stronger winds driven by synoptic features.

    Thank you very much for your comments. As we discussed in the introduction section, the "spectral gap" we investigated in this study is a relatively low spectral

energy range located at frequencies around $10^{-3}$ Hz. The reason for the gap is due to lack of physical processes that could support wind speed fluctuations in this frequency range. The spectral gap represents, to some extent, the transition of atmospheric motion from two-dimensional mesoscale motion to three-dimensional turbulent motion. Many studies have investigated the spectral gap (Van der Hoven, 1957; Fiedler and Panofsky, 1970; Stull, 1988; Kaimal and Finnigan, 1994; Kang and Won, 2016; Larsén et al., 2016), and found different characteristics of the spectral gap in terms of range, spectral density, whether it exists. The spectral gap can be either wide or narrow, and the spectral density of horizontal and vertical wind velocity does not necessarily equal zero.

Regarding whether there is a correlation between two-dimensional mesoscale motion and three-dimensional turbulence. We read the unrelated viewpoint in the literature by Larsén et al. (2016). We are also suspicious of the uncorrelation viewpoint, just as you pointed out, we have observed stronger turbulent spectral density in winter, which is associated with stronger synoptic-scale motions. We have changed the "uncorrelated" statements in the revised manuscript.

2. Equation (1) doesn't really replicate the actual spectra with the coefficients being constant (except for fitting the regions with -2 and -2/3 slopes – which has to be the case, correlation or no correlation. The need for a varying function is demonstrated through use of the Kaimal equations (2) and (3).

Equation (1) is analogy to Lindborg (1999)'s equation, who expressed the spectral structure such that the energy amplitude decreases with wave number as approximately $\kappa^{-5/3}$ (where $\kappa$ is wave number) in the mesoscale range and $\kappa^{-3}$ at lower wave numbers as,
$$S(\kappa) = d_1\kappa^{-5/3} + d_2\kappa^{-3} \tag{R1}$$
where $d_1$ and $d_2$ are constants. Larsén et al. (2013) and Larsén et al. (2016) applied equation (R1) to mid-latitude coastal, offshore, and flat terrain sites, and the weighted spectra $f S(f)$ are expressed as Equation (1).

Equation (1) is verified to describe the mesoscale portion (roughly $10^{-3} < f < 10^{-1}$ Hz) of the frequency range in the full-sale spectral of horizontal wind velocity (Larsén et al., 2013; Kang and Won, 2016; Larsén et al., 2016). Equation (1) also works for this study, in which the horizontal wind spectra are observed at a complex mountainous site. Of course, one has to adjust the coefficients ($a_1$ and $a_2$) in Equation (1) to match the observed wind spectra.

3. The 12-hr peak in the windspeed data is to be expected for any mountain location, and not unique to the QOMS station. One need merely to refer to Chapter 11 of David Whiteman's Mountain Meteorology. (references at the end of this review). Likewise, atmospheric tides produce a 12-hr cycle in the surface winds over at least the tropical ocean.

Thank you very much for your comments and suggestions. We agree that the $12 \text{ hr}^{-1}$ peak in the wind speed spectrum is expected in mountain area and not unique to the QOMS station. However, the $12 \text{ hr}^{-1}$ peak is not always observed in a

horizontal wind spectrum. For example, in this study, the 12 hr$^{-1}$ peak is evident in spring and summer, while the 12 hr-1 peak disappears in winter. Moreover, Kang and Won (2016) investigated the spectral structures of 5-year, 1 min time series of horizontal wind speeds at 100 and 10m heights at the Boulder Atmospheric Observatory tower located in the eastern slope of the Rocky Mountains in USA, but they did not observe the 12 hr$^{-1}$ in the wind spectra. Our intention was to illustrate by the 12 hr$^{-1}$ in wind speed spectrum that the local circulations in the Mt. Everest region is very unique, with significant seasonal differences. However, we did not describe it clearly. We have revised the manuscript and make it more clearly.

In the introduction section, we have added a paragraph to clarify the motivation and purpose of our study on the wind speed spectrum in the Mt. Everest region. Please see the lines 99 to 111 in the revised manuscript: "……*It has long been known that horizontal temperature gradient induced by complex terrain may generate well-defined thermal circulations with distinct diurnal cycle (Whiteman, 2000). A typical mountain-valley circulation system that winds blow up the terrain (upslope and up-valley) during daytime and blow down the terrain (downslope and down-valley) during nighttime. Similar mountain-valley winds circulations are observed in the Himalayas, for example, in the Kali Gandaki Valley in Nepal Himalayas (Egger et al., 2000). However, it is frequently observed that strong up-valley winds persist from noon until sunset and weaker up-valley winds during nighttime, while weak down-valley winds below from sunrise to noon time at a site in the Rongbuk Valley 30 km north of the peak of Mt. Everest (Sun et al., 2007; Sun et al., 2018). Strong down-valley winds are thought to be caused by glacier winds (Sun et al., 2007). However, simulation results indicate that glacier winds do not extend that far and are counteracted by daytime up-valley winds (Cai et al., 2007). Recently, Sun et al. (2018) argued that the temperature gradient between the north and south slopes of the Himalayas allows up-valley winds from the southern slopes to enter the TP, resulting in strong down-valley winds in the northern slopes of the Himalayas. Therefore, the formation process of the strong noon-to-sunset down-valley winds is still debated, and its characteristics and driving mechanisms of formation need more in-depth study……*"

We have also revised the relevant statements in Section 3.2. Please see lines 254 to 259 in the revised manuscript: "……*Interestingly, there is a third peak at the frequency of 12 hr-1. The 12 hr-1 peak in the wind speed spectrum is expected in mountain area, but it is not always observed. For example, Kang and Won (2016) did not observe the 12 hr-1 peak in their wind speed spectra at a site 25 km east of the foothills of the Front Range of the Rocky Mountains. Larsén et al. (2016) did not observe the 12 hr-1 peak either at costal and offshore sites. The 12 hr-1 peak in this study is probably due to the daytime cycle between morning calm wind and afternoon strong southerly wind, which is the result of interactions between the subtropical westerly jet and local valley winds at the QOMS site (Sun et al., 2018)……*"

In the summary and conclusions section, the sentences have been changed to (Lines 395 to 397) "……*In the 15-year full-scale spectra, we identified three peaks*

*at frequencies around 1 yr-1, 1 day-1 and 12 hr-1, respectively. The 12 hr-1 peak in spectra of wind speed is significant in spring and summer, while it disappears in winter. The spectral peak at 12 hr-1 is most likely related to the unique daytime local circulations in the valley where the QOMS station is located……"*

Mountain Locations: There are several examples of mountain-valley, mountain-plains, etc. circulations that go upslope during the day, and downslope during the night. This is a 24-hour cycle in the up-and-downslope component of the wind, but a 12-hour cycle in the wind speed. (This is a classic example of the misleading nature of computing spectra of windspeed instead of dividing the wind into components). Moreover, the authors seem to emphasize only the daytime wind with the 12-hour peak (Line 230). Actually, if there were only a wind peak during the day with no wind change at night, that should show up mostly as a 24-h peak rather than a 12-hr peak. (I see that the observations in Sun et al. (2018) show a nighttime downslope wind peak as well as the daytime peak).

We agree that the mountain wind circulation, which has different systems or types, is a 24-hour cycle in wind direction, but a 12-hour cycle in wind speed. However, under some circumstances, for example, influenced by large-scale circulation or the interaction of multiple local-scale circulations, the diurnal circulation and 12-hour circulation would change. In the Mt. Everest region in this study, the local circulation is not only influenced by mountain winds but also by glacier winds, and even by larger-scale temperature and pressure gradients between the north and south Himalayas.

Figs. 3a and 3c in Sun et al., (2018) show that wind speed and direction exhibit distinct diurnal variation characteristics. Strong south winds build up after about noon. After sunset, the wind speed rapidly decreases and is then replaced by a weak north wind. During the day, after sunrise, the wind speed first increases slightly and then decreases slightly, and it rapidly increases again after noon. During the night, the wind speed barely changes in the monsoon season (Fig. 3a in Sun et al., 2018), while it increases slightly and then decrease in the nonmonsoon season with a much smaller amplitude compared to that in the daytime. This is the reason why we would like to emphasize the daytime wind circulation dominants the 12 hr⁻¹ peak. To avoid misunderstandings, the sentence has been revised to (Line 257 to 259): "……*the 12 hr-1 peak in this study is mainly due to the daytime cycle between morning calm wind and afternoon strong southerly wind, which is the result of interactions between the subtropical westerly jet and local valley winds at the QOMS site (Sun et al., 2018)……"*

Atmospheric Tides. The authors mention (L23 in the abstract and elsewhere) that a 12- hour peak is rarely observed over the ocean. This is untrue. In fair weather over the tropical oceans, the sea-surface temperature doesn't change that much, muting the normal 24-h cycle. Thus, in fair weather, atmospheric tides can be revealed in the surface wind. I have four references, the earlier ones indicating that semidiurnal surface-wind variation was known for over 50 years. The first two are based on

summer data from multiple ships in the 1974 GARP Atlantic Tropical Experiment, GATE, which took place over the tropical Atlantic – unfortunately in hard-to-access papers. The third is from the classic book on atmospheric tides by Chapman and Lindzen. The 4th, a more recent and accessible publication, provides a lot of useful information. While the data differ, all show atmospheric tides show up in the surface wind over the ocean in the same amplitude range.

Jacobs (1980, figure 3) used "all GATE data" (i.e., all ship data) to produce a composite diurnal plot showing semidiurnal wind speed peaks at an average maximum of 4.4 m/s and average minimum of 0.5*(4.2 + 3.9) = 4.05, yielding a range of 0.35 m/s and amplitude of 0.175 m/s.

LeMone (1980, Figure 17)) showed diurnal-average curves, based on seven ships (6 from booms, 1 buoy). The10-m zonal wind has a semidiurnal pattern (2 maxima averaging 2.24 m/s,2 minima averaging1.54 m/s) with a range of 0.7 m/s and an amplitude of 0.35 m/s for 30 August to 20 September. The meridional wind has no semidiurnal cycle.

Chapman and Lindzen (1970, table 2S.9 shows the influence of atmospheric tides on surface winds (amplitude of () = (0.08,0.21) m/s) using data from an island site (Terceira, Azores) and combines data for the whole year.

Ueyama and Deser (2008) annual average amplitude, TAO buoys, ($u$, $v$) = (0.14,0.06) m/s (Fig. 4); June-Nov amplitude: ($u$, $v$) = (0.15,0.07) (Fig. 7) (estimates rough). The Paper also has useful references. This paper also shows a diurnal cycle, comparably small.

Thank you very much for your comments, detailed explanations, and recommended references. We agree with that the 12 hr$^{-1}$ peak is also observed over the ocean. As we reply to your first part of this question, we do not deny that the 12 hr$^{-1}$ peak can be observed, but not always. Our statements in the text were not clear enough, leading to misunderstandings. In the revised manuscript, the sentence in the abstract has been changed to (Line 22 to 23)"……*The 12 hr$^{-1}$ peak is evident in spring and summer but disappears in winter, indicating the seasonal differences in local circulations at the QOMS station……*". Moreover, we also changed the relevant statements in the introduction section, section 3.2, and the summary and conclusions section in the revised manuscript. Please see our response to your first part of this question.

4. The introduction is sloppy, mixing references of observations from fixed points (like the observations here, and those from aircraft, and mixing spectra of wind speed and spectra of the horizontal wind components, and not necessarily covering the range of frequencies of interest.

Thanks for your comments and suggestions. We have revised the introduction and made it more closely aligned with the main theme of the paper. Please see the

fourth paragraph of the introduction section in the revised manuscript.

  Lines 60 to 78: "……. *There are a lot of discussions on the existence of the spectral gap. Many studies observed the a local minimum in the wind speed spectrum at periods of about 1 h and confirmed the gap existence (Fiedler and Panofsky, 1970; Smedman-Högström and Högström, 1975; Gomes and Vickery, 1977; Kaimal and Finnigan, 1994; Vickers and Mahrt, 2003; Yahaya et al., 2003; Kang and Won, 2016; Larsén et al., 2016; Li et al., 2021), although the gap was not as significant as that of Van der Hoven (1957). However, the spectral gap is not always well defined. Based on the assumption of the spectral gap, we might expect there to be little variability in the wind speed on these time scales, and that in certain atmospheric conditions the gap may not exist at all (LeMone, 1976; Gjerstad et al., 1995; Heggem et al., 1998). Studies question the existence of the gap in Van der Hoven (1957)'s wind speed spectrum mainly because the high-frequency region was observed during the passage of a hurricane, which increases the spectral density in the high-frequency range compared to standard atmospheric conditions (Smedman-Högström and Högström, 1975; Kang and Won, 2016; Larsén et al., 2016). Multiple atmospheric processes are reported that contribute to the generation of variations in the spectral gap region, including large cumulus clouds (Stull, 1988), convective cells (Gjerstad et al., 1995; Heggem et al., 1998), and horizontal roll vortices (Heggem et al., 1998). Larsén et al. (2016) suggested that the gap is jointly regulated by the variation in horizontal wind variations from the two-dimensional mesoscale motions and three-dimensional boundary layer turbulence, and may be visible or invisible depending on the relative contribution to the fluctuation from the microscale and mesoscale motions. Moreover, the presence or absence of a spectral gap also depends on the sampling strategy. The spectral gap would be evident and big enough if the data is observed from a fixed location and only along the wind direction. However, if the data is measured perpendicular to the roll vortices, for example, in an aircraft flying normal to the rolls, a single spectral peak indicating the combination of rolls and 3D large convective eddies will be obtained, and the spectral gap will vanish…….*"*

For example, the Sun and Lenschow paper cited uses aircraft data, which weren't necessarily sampled parallel to the wind. Second, they compute spectra of the wind components rather than speed. Third, their frequency range is severely limited by the length of the flight tracks and thus does not sample the synoptic (2-D) scale. Rather, their lower frequencies, in TOGA COARE at least, are likely associated with precipitating convection. Thus, their results have little to do with the theme of this paper. Focusing only on the papers dealing with the gap between synoptic and turbulent motions would greatly shorten and improve the introduction.

  Thanks for your suggestions. According to your suggestion, we have removed Lenschow and Sun's work from the introduction section in the revised manuscript.

Since the current introduction deals with spectral gaps that involve only boundary-layer motions, it should be noted that the presence or absence of a gap is a function

of the sampling strategy.

That is, if you are sampling from a point (or fixed tower), you will likely get a two-peak spectrum in the convective boundary layer in the presence of horizontal convective rolls (which are nearly parallel to the wind and take of the order of 30 min to an hour to pass a point on the surface) that coexist with large 3-d convective eddies (which pass a point in more like 5-10 min). This separation in periods is useful for separating out the two structures for analysis, even though the structures of roughly the same scale in the crosswind direction. With only along-wind data, one might think that the spectral gap is big enough for zero nonlinear terms under Charnock's criterion.

However, if you sample using an aircraft flying normal to the roll axis, you get a single spectral peak, corresponding to the roll/convective cell combination. As can be seen in clear-air radar echoes in the presence of rolls, the convective cells (large eddies) lie in the rolls' upwelling portion – i.e., they are tightly correlated. So, even if you get the spectral gap identified by Charnock as suggesting zero nonlinear terms with the alongwind data, the crosswind data tell another story.

*The above three paragraphs address the same issue, and we will respond to them together here.*

*According to your suggestion, we included the discussion on the presence or absence of a spectral gap is related to the sampling strategy in the revised manuscript. Please see sentences in the fourth paragraph of the introduction section: "……Moreover, the presence or absence of a spectral gap also depends on the sampling strategy. The spectral gap would be evident and big enough if the data is observed from a fixed location and only along the wind direction. However, if the data is measured perpendicular to the roll vortices, for example, in an aircraft flying normal to the rolls, a single spectral peak indicating the combination of rolls and 3D large convective eddies will be obtained, and the spectral gap will vanish……"*

Thus, unlike what is stated on line 72, a spectral gap does NOT mean that motions are weakly correlated. In fact, they are strongly correlated and interact in this case. Clearly, the presence of the convective cells in the upwelling portion of the rolls indicates a correlation. A paper you cite, LeMone (1976) discusses the interaction of the rolls and cells.

*We agree with your that the statement mesoscale and microscale motions are weakly or uncorrelated is not correct. As the reviewer pointed out, in figure 6 in this study, stronger turbulence results from stronger winds driven by synoptic events. This phenomenon indicates clearly the correlation between mesoscale motions and microscale turbulences. We have deleted the uncorrected statement in the entire manuscript and pointed this out in the conclusion section: "……However, it does not mean that the 3D turbulence and the 2D mesoscale motions are not or weakly correlated……"*

5. Moreover, I don't think that a list of spectral peaks at different geographic locations is illuminating unless they are carefully grouped according to terrain and sampling strategy (length, seasons, fixed vs traveling platform and whether/how its track is related to wind direction).

Thanks for your comments and suggestions. In this study, we do not simply report the observed 12-hour and 24-hour spectral peaks, as well as the spectral gap at the QOMS site. As we wrote in the introduction section and the responses to your comments, the local circulation observed at the QOMS site is not a typical mountain-valley circulation system, nor a typical glacier circulation system. The formation processes of the strong noon-to-sunset down-valley winds are still not very clear. The spectral characteristics in this study, which across different scales, can help us, to some extent, better understand the formation mechanisms of the local circulations on the north slope of Mt. Everest.

We analyzed our data in different seasons, but we only have measurements at multiple fixed points, do not have aircraft or drone observations. In the future, we may conduct aircraft or drone experiments, and establish observations along and across the valley, to have a comprehensive picture of the local circulations.

**Specific comments:**

L65. Lenschow and Sun applies to aircraft data, not data collected along the wind. Also, they show spectra of the wind components, which is different from what is done here. Should check these studies to see if they correspond to what is done in this paper.

Leschow and Sun's work does not fit to what is done in this study and the citation has been deleted.

In the introduction, the discussion is longer than it needs to be. What is of interest here is the gap between synoptic (essentially 2-D) and turbulence (3-D) motions, rather than a gap between structures in the boundary layer.

Thanks for your comments and suggestions. We revised the introduction and focused more on the gap between 2D mesoscale and 3D turbulent motions. Moreover, we also revised the inaccurate statement "uncorrelation between mesoscale motion and turbulent motions".

Please see the fourth paragraph of the introduction in the revised manuscript (Lines 60-78): "……*There are a lot of discussions on the existence of the spectral gap. Many studies observed the a local minimum in the wind speed spectrum at periods of about 1 h and confirmed the gap existence (Fiedler and Panofsky, 1970; Smedman-Högström and Högström, 1975; Gomes and Vickery, 1977; Kaimal and Finnigan, 1994; Vickers and Mahrt, 2003; Yahaya et al., 2003; Kang and Won, 2016; Larsén et al., 2016; Li et al., 2021), although the gap was not as significant as that of Van der Hoven (1957). However, the spectral gap is not always well defined. Based on the assumption of the spectral gap, we might expect there to be little variability in the wind speed on these time scales, and that in certain atmospheric conditions the gap may not exist at all (LeMone, 1976; Gjerstad et al., 1995; Heggem et al., 1998). Studies question the*

*existence of the gap in Van der Hoven (1957)'s wind speed spectrum mainly because the high-frequency region was observed during the passage of a hurricane, which increases the spectral density in the high-frequency range compared to standard atmospheric conditions (Smedman-Högström and Högström, 1975; Kang and Won, 2016; Larsén et al., 2016). Multiple atmospheric processes are reported that contribute to the generation of variations in the spectral gap region, including large cumulus clouds (Stull, 1988), convective cells (Gjerstad et al., 1995; Heggem et al., 1998), and horizontal roll vortices (Heggem et al., 1998). Larsén et al. (2016) suggested that the gap is jointly regulated by the variation in horizontal wind variations from the two-dimensional mesoscale motions and three-dimensional boundary layer turbulence, and may be visible or invisible depending on the relative contribution to the fluctuation from the microscale and mesoscale motions. Moreover, the presence or absence of a spectral gap also depends on the sampling strategy. The spectral gap would be evident and big enough if the data is observed from a fixed location and only along the wind direction. However, if the data is measured perpendicular to the roll vortices, for example, in an aircraft flying normal to the rolls, a single spectral peak indicating the combination of rolls and 3D large convective eddies will be obtained, and the spectral gap will vanish……"*

L82-84. Height dependence depends on the variable observed. With rolls as an example, the spectral gap in vertical velocity would become MORE evident with height. Of course, the horizontal winds vary more near the surface. (don't understand why you need a reference for this – it's common sense).

The references have been deleted in the revised manuscript.

L90. Studying a map of Beijing, a 12-hour peak in wind speed seems likely, associated with the 24-hour upslope-downslope wind cycle with nearby higher terrain.

Thanks for your comments. We have added the following sentence in the revised manuscript: "……*The 1/2 day spectral peak observed in Beijing is probably related to the diurnal cycle of upslope and downslope winds, as there are mountains to its west and north……*".

Figure 1. A terrain map would be more helpful. It appears that the site isn't on a slope but is in a valley. Details are important – see David Whiteman's Mountain Meteorology (reference below). Also the valley location would make the comparison with the wind rose more meaningful.

If I get it correctly, what you referred to as the "terrain map" should be the DEM map. The DEM map was used, like what is shown in Fig. R1. However, on the DEM map, the glacier and land surface contrasts are not visible. The current map could show clearly the topography of the Mt. Everest region and also show the valley where the QOMS site is located. Thus, we did not change the map.

The QOMS site is located in the valley, not on slope. Moreover, a more detailed description of the site has been added in the revised manuscript: "……*The QOMS station is located in the Rongbuk Valley, at an elevation of 4276 m above sea level, and*

*is approximately 30 km north of the peak of Mt. Everest (Figure 1). The QOMS is located at the bottom of the Rongbuk Valley, which runs in a northeast-southwest direction (approximately 10° to 190°), and the width of the valley at the site is about 1.5 kilometers. The ground surface around QOMS is flat and is covered with sand, gravel, and sparse short grass……"*

[Figure]

Fig. R1: Topography of the Mount Everest region.

L118. 30 km north of the peak of Mt. Everest in a valley is a significantly better description than "on the north slope of Mt. Everest."

Thanks for your comments and suggestions. The sentence has been changed to "……*The QOMS station is located in the Rongbuk Valley, at an elevation of 4276 m above sea level, and is approximately 30 km north of the peak of Mt. Everest……*" in the revised manuscript.

L160-165. A reasonable summary of the wind roses would start with a statement about how much the winds are along the valley axis – I was surprised that this comparison isn't made until farther down in the paper and that it was not mentioned in the conclusions.

Thanks for your suggestions. We have added sentences clarifying how much winds are along the valley in the revised manuscript: "……*The compass roses with 16 wind directions show that about 71.5% of the winds follow the direction of the river valley, with northerly and southerly winds being the predominant wind directions. Specifically, at the height of 1.5 m, northerly to northeasterly winds account for 28.7%, while southeasterly to southwesterly winds make up 42.8%......*". Moreover, we also included

this statement in the conclusion section: "……*About 71.5 % of the winds blow in the direction of the river valleys at the QOMS station……*"

L193. Delete "changing."

"changing" has been deleted.

L195-6. Are the increasing near-surface wind speeds over flatter terrain?

Both of the studies cited here, they used wind data collected at meteorological stations of China Meteorological Administration. All meteorological stations are built on flat ground, and the surroundings are generally open and unobstructed. Thus, the increase in near-surface wind speed is over flatter terrain compared to the condition around the QOMS station.

L225. This is hardly surprising and well-known that the diurnal cycle is stronger over land. Are references even needed? (i.e., this is discussed in most introductory meteorology books).

Thanks for your comments and suggestions. The references have been deleted in the revised manuscript.

L230. Don't Sun et al. see the classic mountain-valley wind, with the daytime wind influenced by the westerly jet and a second night-time maximum associated with downslope wind? You have to consider the night-time wind as well. If there was only a maximum in the 12-hour period during the day influenced by the subtropical westerly jet, that would be a 24-hour cycle. (see general comments)

A typical mountain-valley circulation system that winds blow up the terrain (upslope and up-valley) during daytime and blow down the terrain (downslope and down-valley) during nighttime. However, Sun et al. (2018) and Sun et al. (2007) observed that strong up-valley winds persist from noon until sunset and weaker up-valley winds during nighttime, while weak down-valley winds below from sunrise to noon time at a site in the Rongbuk Valley 30 km north of the peak of Mt. Everest. Thus, the circulation system at the QOMS station is completely different from the mountain-valley circulation system.

As our response to your third general comment, Figs. 3a and 3c in Sun et al., (2018) show that wind speed and direction exhibit distinct diurnal variation characteristics. Strong south winds build up after about noon. After sunset, the wind speed rapidly decreases and is then replaced by a weak north wind. During the day, after sunrise, the wind speed first increases slightly and then decreases slightly, and it rapidly increases again after noon. During the night, the wind speed barely changes in the monsoon season (Fig. 3a in Sun et al., 2018), while it increases slightly and then decrease in the nonmonsoon season with a much smaller amplitude compared to that in the daytime. This is the reason why we would like to emphasize the daytime wind circulation dominants the 12 hr-1 peak. To avoid misunderstandings, the sentence has been revised to (Line 257-259): "……*the 12 hr-1 peak in this study is mainly due to the daytime cycle between morning calm wind and afternoon strong southerly wind, which is the*

*result of interactions between the subtropical westerly jet and local valley winds at the QOMS site (Sun et al., 2018)……*"

L236-8. Not necessarily. Perhaps they would be close because the average winds were close.

Thanks for your comments. We agree that the similarity of daily spectra in 2015 and 2016 calculated from the 10-Hz sonic data is probably due to the averaged wind speed are close. Indeed, the one-year averaged daily wind speeds are very close to each other. Moreover, we could see the annual average wind speed shows little variation (Figure 4b). Thus, we believe that using the wind speed spectra calculated from 2015 and 2016 as a climatological representative of the daily wind spectra is reasonable.

L253. As expected, the mountain-valley system is stronger in the summer, and it is not surprising that a 12-h peak is stronger than the 24-h peak in wind SPEED. For the wind component along the slope, the period would be 24 h. This is an example of the misleading aspect of using speed instead of wind components.

We agree that the 12-h and 24-h peaks are associated with the local circulation systems (mountain-valley and glacier wind systems), which are more pounced in summer and weaker in winter. The 12-hour peak is greater than the 24-hour peak, and we believe that the primary reason is the significant variation in wind speed during the day time (See more details in our response to your third general comment).

L270. Regarding Kang and Won, what was the length of the dataset used?

In Kang and Won (2016)'s work, they used a 5-year time series of horizontal wind data measured by prop-vane anemometers.

A cautionary note: The relationship between scale in m and frequency in Hz is often a function of the windspeed (if features are carried by the wind). For winter, and stronger winds, the same scale of eddy (thinking about km-scale) will produce a higher frequency. For winter, that eddy might be smaller since the boundary layer could be smaller than in the summer.

Thanks for your comments and suggestions. It is a very interesting point and we added a sentence in the revised manuscript (Line 309-310): "……*Moreover, stronger winds would produce higher frequencies as the relationship between scale in m and frequency in Hz is often a function of the windspeed……*"

Eq (1) doesn't make much sense unless the coefficients are functions. Without functions, how can it match any observations "perfectly" (L312).

We agree that the coefficients in Eq. (1) are variable to match observations with different power spectral densities. The meaning of Eq. (1) is that the energy amplitude decreases with frequency as approximately $f^{5/3}$ in the mesoscale range, and as $f^3$ at higher frequencies. In this study, it is obvious that Eq. (1) describes only a portion of the suggested frequency range in the full-scale spectral of wind speed observed at the QOMS station. In fact, the second term in the right-hand side of Eq. (1) becomes less

significant with increasing frequency and the spectrum simply converges to $f^{2/3}$.

Figure 8. and Eq (1). The line labeled "Eq. (1)" is only the low-frequency part, along with Kaimal and Finnigan's spectral function at the higher frequencies. This does not show that Eq. (1) works if the coefficients are constants. And it is curious that you don't extend Kaimal's spectral power to the higher frequencies.

As the response to your previous question, coefficients in Eq. (1) are variable to match observations with different power spectral densities. Regarding the Kaimal spectrum, we agree it is can be extended for higher frequencies. Kaimal spectra were derived from the Kansas experiment (Kaimal et al., 1972), where is relatively flat and homogeneous. In this study, Kaimal spectrum is not extendible to higher frequencies. We speculate that the main reason is the highly complex topography and underlying surface characteristics around the QOMS site in the Mt. Everest region. Kaimal and Finnigan (1994) also reported that when flow passes over varying terrain and hills, the velocity spectrum in the high-frequency range changes. This is due to the greater contribution to kinetic energy from mechanically (shear) produced turbulence than we expected in a classical convective mixed layer, which leads to the increase in spectral density in the high-frequency range. The wind spectra over complex terrain and heterogeneous surfaces in the Mt. Everest region is a very interesting topic. In this study, we focus on the spectral gap in the transition range between mesoscale and microscale. In the future, we will further study the applicability of the Kaimal spectrum in the microscale spectral range in this region.

L328. It does appear that you can linearly composite the low- and high-frequency data, but this does not imply that the two are independent of one another.

Thanks for your comments. To clarify, the sentence has been revised to "……*In this study, in line with Larsen et al. (2016)'s hypothesis, microscale and mesoscale wind spectra are linearly composited, however, it does not mean that the 3D turbulence and the 2D mesoscale motions are independent from each other……*"

Figures 5-8. The rising spectral density at the high frequency end reflects the presence of white noise and should be eliminated or at least pointed out.

Thanks for your suggestions. We have pointed out this issue at the end of section 3.2 in the revised manuscript: "……*Note that the increase in spectral density at the high frequency end of the daily wind spectra (dotted and dashed green curves) is due to the presence of white noise……*"

Figure 8. Did you evaluate periods for stable (L>0) and unstable (L<0) separately in Eq. (2)- (3) and then combine? There should be a more complete explanation about how the calculation is done. Also, more precise to say 'added' rather than 'merged."

Yes, the dissipation rate of turbulent kinetic energy and wind spectra were calculated for stable ($z/L{\geq}0$) and unstable ($z/L{\leq}0$) conditions seperately and then combined. We have added sentences in the revised manuscript for a better explanation about how the calculation was done. Please see lines 374-376 in the revised

manuscript: "……*The friction velocity (u_*) and Obukhov length (L) were calculated from the eddy-covariance data. The dissipation rate of turbulent kinetic energy (ϕ_ε) and spectra of horizontal wind velocity (S_V (f)) were calculated for stable (z/L≥0) and unstable (z/L≤0) conditions and then combined……*".

Moreover, the "merged" has been changed to "added" in the revised manuscript (See Figure 8).

6. Summary and conclusions:

L342. Again, misleading to write "on the north slope of Mt. Everest," since the location of the station in the valley has strong control over the wind direction.

Thanks for your comments and suggestions. The sentence has been changed to "……*A 15-year time series of horizontal wind speeds observed at a mountainous site in a valley 30 km north of the peak of Mt. Everest was used to investigate the full-scale spectra of horizontal wind speed……*" in the revised manuscript.

L346. Along the valley! (and not north-south)

Thanks for your comments. The sentence has been changed to "……*About 71.5 % of the winds blow along the direction of the river valleys at the QOMS station……*" in the revised manuscript.

L353-4. Not a result of this paper.

The sentences have been deleted in the revised manuscript.

L354-5. More accurate to say that the 12-h peak is common in spectra of wind SPEED. This is an important distinction. "Some urban sites" -- are these urban sites close to/in mountains, like Hong Kong or Beijing? Again, not a result of this paper.

Then sentence has been changed to "……*The 12 hr⁻¹ peak in the wind speed spectra is significant in spring and summer, while it disappears in winter……*", and results not of this paper have been deleted.

L356. The 12-h period is not unique; nor is it related to the single peak during the day, as explained earlier.

Thanks for your comments. We still believe the wind variation in the daytime is the main driver of the 12-h peak. We revised the sentence to "……*The spectral peak at 12 hr⁻¹ is most likely related to the daytime local circulations in the valley where the QOMS station is located……*"

L363. That's only 16.7 minutes, well within convective boundary layer frequencies! If you look at the curves carefully, the minima tend to be at lower frequencies, which makes more sense. You have a better estimate in the body of the paper. (L298)

Thanks for your comments and suggestions. The location and intensity of the spectral gap are different between summer and winter. We changed the sentence to "…..*The spectral gap is observed between the frequency of $1.0\times10^{-4}$ and $1.0\times10^{-3}$*

*Hz*……" in the revised manuscript.

L366-367. For the many reasons (and based on the results) the synoptic and turbulence spectra ARE correlated.

*Thanks for your comments. The sentence has been changed to "……Larsén et al. (2016) hypothesized that the spectra of 3D microscale turbulence and 2D mesoscale wind variations are linearly composited……" in the revised manuscript.*

REFERENCES

Chapman, S., and R. S. Lindzen, 1969: Atmospheric Tides. D. Reidel Publishing Company, the Netherlands, 200 pp.

Jacobs, C.A., 1980: Mean diurnal and shorter period variations in the air-sea fluxes and related parameters during GATE, In Siedler, Gerold, and John D. Woods, 1980: Oceanography and Surface Layer Meteorology of the B/C-Scale, GATE, V1, Pergamon press, 294 pp. Supplement 1 to Deep-Sea Research Part A, v. 26, pp 65-98. https://doi.org/10.1016/B978-1-4832-8366-1.50009-1.

LeMone, M. A., 1980: The marine boundary layer, pp. 183-231, in Workshop on the Planetary Boundary Layer. J.C. Wyngaard, ed. American Meteorological Society, 322 pp.

Ueyama, R. and Deser, C., 2008. A climatology of diurnal and semidiurnal surface wind variations over the tropical Pacific Ocean based on the tropical atmosphere ocean moored buoy array. Journal of climate, 21(4), pp.593-607.

Whiteman, C. D., 2000: Mountain Meteorology: Fundamentals and Applications. Oxford University Press.355 pp.

References:

Fiedler, F. and Panofsky, H. A.: Atmospheric scales and spectral gaps, Bulletin of the American Meteorological Society, 51, 1114-1120, https://doi.org/10.1175/1520-0477(1970)051<1114:ASASG>2.0.CO;2, 1970.

Gjerstad, J., Aasen, S. E., Andersson, H. I., Brevik, I., and Løvseth, J.: An analysis of low-frequency maritime atmospheric turbulence, Journal of Atmospheric Sciences, 52, 2663-2669, https://doi.org/10.1175/1520-0469(1995)052<2663:AAOLFM>2.0.CO;2, 1995.

Gomes, L. and Vickery, B. J.: On the prediction of extreme wind speeds from the parent distribution, Journal of Wind Engineering and Industrial Aerodynamics, 2, 21-36, https://doi.org/10.1016/0167-6105(77)90003-4, 1977.

Heggem, T., Lende, R., and Løvseth, J.: Analysis of long time series of coastal wind, Journal of the Atmospheric Sciences, 55, 2907-2917, https://doi.org/10.1175/1520-0469(1998)055<2907:AOLTSO>2.0.CO;2, 1998.

Kaimal, J. C., Wyngaard, J. C., Izumi, Y., and Coté, O. R.: Spectral characteristics of surface-layer turbulence, Quarterly Journal of the Royal Meteorological Society, 98, 563-589, https://doi.org/10.1002/qj.49709841707, 1972.

Kaimal, J. C. and Finnigan, J. J.: Atmospheric boundary layer flows: Their structure and measurement, Oxford University Press, New York, 1994.

Kang, S.-L. and Won, H.: Spectral structure of 5 year time series of horizontal wind speed at the Boulder Atmospheric Observatory, Journal of Geophysical Research: Atmospheres, 121, 11,911-946,967, https://doi.org/10.1002/2016JD025289, 2016.

Larsén, X. G., Vincent, C., and Larsen, S.: Spectral structure of mesoscale winds over the water, Quarterly Journal of the Royal Meteorological Society, 139, 685-700, https://doi.org/10.1002/qj.2003, 2013.

Larsén, X. G., Larsen, S. E., and Petersen, E. L.: Full-scale spectrum of boundary-layer winds, Boundary-Layer Meteorology, 159, 349-371, https://doi.org/10.1007/s10546-016-0129-x, 2016.

LeMone, M. A.: Modulation of turbulence energy by longitudinal rolls in an unstable planetary boundary layer, Journal of Atmospheric Sciences, 33, 1308-1320, https://doi.org/10.1175/1520-0469(1976)033<1308:MOTEBL>2.0.CO;2, 1976.

Li, B., Li, C., Yang, Q., Tian, Y., and Zhang, X.: Full-scale wind speed spectra of 5 Year time series in urban boundary layer observed on a 325 m meteorological tower, Journal of Wind Engineering and Industrial Aerodynamics, 218, 104791, https://doi.org/10.1016/j.jweia.2021.104791, 2021.

Lindborg, E.: Can the atmospheric kinetic energy spectrum be explained by two-dimensional turbulence?, Journal of Fluid Mechanics, 388, 259-288, https://doi.org/10.1017/S0022112099004851, 1999.

Smedman-Högström, A.-S. and Högström, U.: Spectral gap in surface-layer measurements, Journal of Atmospheric Sciences, 32, 340-350, https://doi.org/10.1175/1520-0469(1975)032<0340:SGISLM>2.0.CO;2, 1975.

Stull, R. B.: An introduction to boundary layer meteorology, Kluwer Academic Publishers, Dordrecht, https://doi.org/10.1007/978-94-009-3027-8, 1988.

Sun, F., Ma, Y., Li, M., Ma, W., Tian, H., and Metzger, S.: Boundary layer effects above a Himalayan valley near Mount Everest, Geophysical Research Letters, 34, https://doi.org/10.1029/2007GL029484, 2007.

Sun, F., Ma, Y., Hu, Z., Li, M., Tartari, G., Salerno, F., Gerken, T., Bonasoni, P., Cristofanelli, P., and Vuillermoz, E.: Mechanism of daytime strong winds on the northern slopes of Himalayas, near Mount Everest: Observation and simulation, Journal of Applied Meteorology and Climatology, 57, 255-272, https://doi.org/10.1175/JAMC-D-16-0409.1, 2018.

Van der Hoven, I.: Power spectrum of horizontal wind speed in the frequency range from 0.0007 to 900 cycles per hour, Journal of Atmospheric Sciences, 14, 160-164, https://doi.org/10.1175/1520-0469(1957)014<0160:PSOHWS>2.0.CO;2, 1957.

Vickers, D. and Mahrt, L.: The cospectral gap and turbulent flux calculations, Journal of Atmospheric and Oceanic Technology, 20, 660-672, https://doi.org/10.1175/1520-0426(2003)20<660:TCGATF>2.0.CO;2, 2003.

Yahaya, S., Frangi, J. P., and Richard, D. C.: Turbulent characteristics of a semiarid atmospheric surface layer from cup anemometers – effects of soil tillage treatment (Northern Spain),

Ann. Geophys., 21, 2119-2131, 10.5194/angeo-21-2119-2003, 2003.